# Label-free concurrent 5-modal microscopy (Co5M) resolves unknown spatio-temporal processes in wound healing

Markus Seeger[1,2], Christoph Dehner [1,2], Dominik Jüstel[1,2,3] & Vasilis Ntziachristos [1,2 ✉]

The non-invasive investigation of multiple biological processes remains a methodological challenge as it requires capturing different contrast mechanisms, usually not available with any single modality. Intravital microscopy has played a key role in dynamically studying biological morphology and function, but it is generally limited to resolving a small number of contrasts, typically generated by the use of transgenic labels, disturbing the biological system. We introduce concurrent 5-modal microscopy (Co5M), illustrating a new concept for label-free in vivo observations by simultaneously capturing optoacoustic, two-photon excitation fluorescence, second and third harmonic generation, and brightfield contrast. We apply Co5M to non-invasively visualize multiple wound healing biomarkers and quantitatively monitor a number of processes and features, including longitudinal changes in wound shape, microvascular and collagen density, vessel size and fractality, and the plasticity of sebaceous glands. Analysis of these parameters offers unique insights into the interplay of wound closure, vasodilation, angiogenesis, skin contracture, and epithelial reformation in space and time, inaccessible by other methods. Co5M challenges the conventional concept of biological observation by yielding multiple simultaneous parameters of pathophysiological processes in a label-free mode.

[1] Chair of Biological Imaging, Central Institute for Translational Cancer Research (TranslaTUM), School of Medicine, Technical University of Munich, Munich, Germany. [2] Institute of Biological and Medical Imaging, Helmholtz Zentrum München, Neuherberg, Germany. [3] Institute of Computational Biology, Helmholtz Zentrum München, Neuherberg, Germany. ✉email: bioimaging.translatum@tum.de

Wounds compromise the skin's protective, preservative, and sensory functions[1]. The need to unravel the wound healing process in detail has been well recognized as a means to better understand the underlying processes that drive or impede healing and to assess new therapies[1–5]. However, extensive and comprehensive investigations of the central components involved in tissue repair are hindered by a lack of appropriate methods that can longitudinally access intertwined biological processes[5,6]. Wound healing involves skin components that are characterized by functional units, including microvessels, collagen fibrils, keratinocytes, and sebaceous glands, that respond in an interdependent series of events over time and space to realize tissue repair[7]. The spatial and temporal characteristics of the healing bio-hallmarks, e.g., neovascularization, connective tissue restructuring, or stem cell activation, subdivide the healing process into four essential phases[1,3,7–18]: hemostasis, inflammation, proliferation, and remodeling; and into three zones[18,19]: former wound site, leading edge, and proliferative hub. The spatial extent, temporal duration, and interplay of these four phases and three zones, collectively depictable as a schematic wound healing model, yield information about an injury's severity, the health of the tissue, and the quality of recovery[20,21].

It is particularly important to elucidate the longitudinal responses of the tissue layers[1,9,14], functional units and skin appendages during healing[1,7,9,14,18] to resolve their mutual interdependency. Critical unknown orchestrations play a role in the interplay between misregulated stem cell activation and vasculopathies (e.g., abnormal anastomoses[22] or microangiopathy[23–25]) and associated impaired epithelial growth and imbalanced tissue matrix maintenance (e.g., diabetic ulcers[5] or scar formations[26,27]). It is currently challenging to unravel these effects and their association with underlying conditions, such as diabetes, sepsis, or infections: no method can reveal sufficient wealth of contrast to concurrently capture the disease-specific entanglement of bio-hallmarks involved in the healing process and lead to an accurate wound healing model.

In-depth understanding of wound healing thus requires dynamic and concurrent imaging of multiple biological contrast at micrometer precision, over large fields of view (several mm$^2$), and over long time periods (several weeks), in order to formulate accurate wound healing models to evaluate the impacts of e.g., disease and wound treatment on the healing process. Optical imaging methods have been considered for the non-invasive investigation of healing bio-hallmarks in spatial and temporal terms[1–4,11–14,21,22,28–33]. Laser Doppler imaging, optoacoustic microscopy, and optical coherence tomography have been deployed to selectively monitor the microcapillary vasculature during wound healing[3,11,22,34–37]. Wound-induced vasculature changes were by this means observed as neovascularization ~1 week after injury[11,22,34,36,37], as increases in tortuosity at the leading edge[3,11,36–38], and as vasodilation at ~1–2 week post injury in the proliferative zone[3,34,36,37], but were not linked to their associated regulation by activated stem cells. Likewise, multiphoton microscopy, Raman microscopy, and spectral optical imaging have been considered for resolving changes in the connective tissue after injury and the reorganization of the extracellular collagen matrix[1–3,11,18,21,27,31,32,39,40]. Responses of the connective tissue to wounds were in this way described as skin contracture at ~2 weeks after injury[11] and sealing of the wound site[11] at ~3–5 days[32] up to 400 μm away from the wound[15], but were not related to the presupposed surge in nutrient supply provided by increased blood flow. Intravital microscopy has been also employed to resolve stem and epithelial cells as well as sebaceous glands[11,18]. By that, stem and epithelial cell activation as well as regeneration of sebaceous glands were found at ~1–2 weeks after injury[11] in the proliferation zone (0.5–1.5 mm)[19], and sebaceous glands migration towards wound after 10 days on across an area ~1.5 mm away from the wound[18], but did not correlate these processes with changes in the extracellular collagen matrix that modulates cell mobility in skin. Whereas each of these methods captured a partial view of the healing wound, monitoring of a single or few healing bio-hallmarks resulted in contradictory and incomplete findings. Independently performed studies using different animal models and imaging protocols further made the merger of the findings questionable if not impossible.

Multimodal microscopy has been suggested to overcome limitations of single-modal microscopy by enlarging the number bio-hallmarks monitored[1,2,21,32,41–43] and was so far shown to enable simultaneous imaging of angiogenic and metabolic changes, collagen synthesis, and wound closure. However, previous multimodal studies of wound healing utilized labels that may disturb the specimen[32,42], stitched adjacent FOVs affecting the spatial analysis[42,43], captured small FOVs[2,21,32,41] or a fraction of the healing course after wound infliction[21,32,43], or suffered from poor or unequal resolution and co-registration of the modalities[2,41]. While these studies extracted quantitative information of the imaged bio-hallmarks, the yielded wound healing models simplified the analysis by dividing the affected tissue area in separated zones, by describing the healing events separately, by missing remote aspects at far distances or late time points, or by omitting an equalized mathematical description of the hallmark's processes. Thus, no method allows to concurrently capture a sufficient number of bio-hallmarks of healing wounds and to observe, analyze, and quantify the healing process. Subsequently, no accurate and easy-to-grasp wound healing model capturing the healing bio-hallmarks' spatio-temporal entanglement was yet achieved, emphasizing the need for a new investigation strategy to probe the role and interactions of the central components of healing and aberrations caused by disease.

To address the aforementioned limitations of methods for in vivo biological observation, we build upon the postulation that label-free and concurrent imaging of multiple targets can offer new insights and comprehensive understanding of biological processes, without altering the biological system with invasive approaches or transgenic modifications. To empower multimodal microscopy and deliver an observation tool in biological interrogations, we re-engineered a concurrent five-modal microscope (Co5M) that features now (Fig. 1) at least 20-fold faster imaging, improved image quality with 12 dB-higher SNR, and 3-fold more precise spatial co-registration over previous implementations[44–52]. The five modes comprise optical-resolution optoacoustic microscopy (OAM) for imaging microvasculature, two-photon excitation fluorescence (2PEF) modality for capturing skin morphology based on skin autofluorescence, second and third harmonic generation (SHG and THG) modes for visualizing the extracellular collagen matrix and lipids/sebaceous glands, respectively, and brightfield (BF) microscopy for providing an anatomical reference. For the herein used athymic nude Hsd Foxn1 mouse model, we expect Co5M's OA signals to be dominated by hemoglobin[53–57], SHG signals by the extracellular collagen matrix[21,58–62], THG signals by lipid bilayers (cell membranes, especially keratinocytes of the epidermis) and local lipid accumulations such as in sebaceous glands[21,58–61], and 2PEF signals by elastin, FAD, NADH, and keratin[21,59,62,63]. Co5M's modalities access that by intrinsic biological contrast, which is analyzed to outline the spatial characteristics of the observed biological moieties. The synchronization of all equipped devices (i.e., lasers, detectors, scanning units, and data acquisition cards) by a system-wide shared triggering scheme and a time-locked data acquisition allows simultaneous recording of the modalities with negligible crosstalk. In combination with a fully

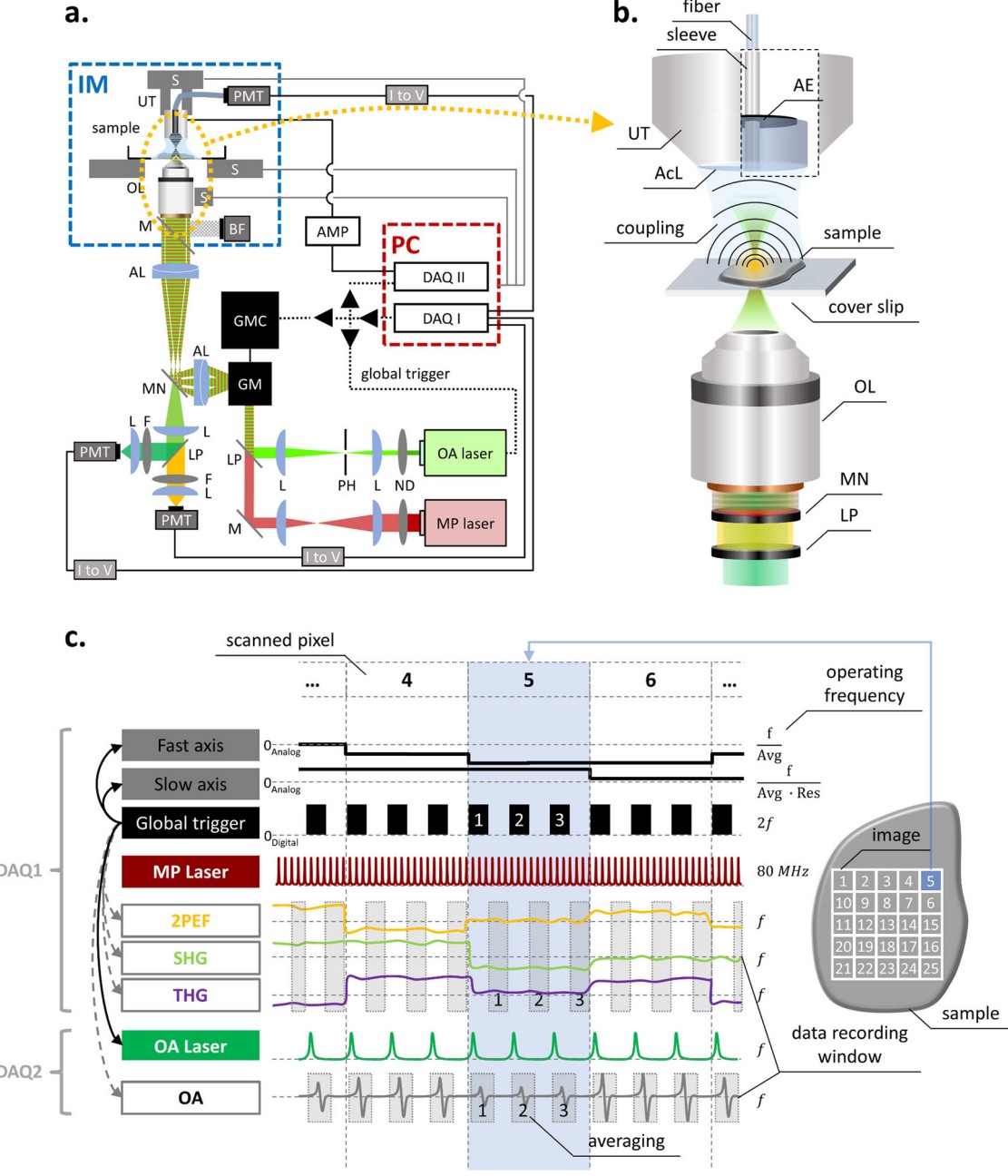

**Fig. 1 Merged Co5M system integrating MP and OR-OAM subsystems. a** Co5M integrates multiphoton and optical-resolution optoacoustic modalities along with brightfield microscopy for referencing purposes. **b** Close-up depiction of the interrogation area for simultaneous multimodal acquisition. Abbreviations: AcL acoustic lens, AE active element, AL achromatic doublet lens, AMP low noise amplifier, BF brightfield camera, DAQ data acquisition card, F optical filter, GM galvanometric mirror scanner, GMC GM controller, IM inverted microscope, L plan-convex lens, LP longpass dichroic, M dielectric mirror, MN multinotch dichroic, ND neutral density filter, OL microscope objective lens, PH pinhole, PMT photomultiplier tube, S high-precision motorized stage, UT ultrasound transducer. **c** Data acquisition scheme based on two DAQs electrically synchronized by a global trigger. DAQ1 is controlling for the slow and fast scanning axis of the galvanometric mirrors, provides the global trigger, and records MP signals at the off-states of the global trigger. DAQ2 is recording the OA signals induced by the actively triggered OA laser in a streaming-like acquisition mode.

automated scanning procedure based on fusing laser- and sample-scanning concepts, Co5M can now obtain high-fidelity images of key tissue components with micrometer resolution across millimeter-sized areas within preclinical time-frames, enabling the extraction of quantitative metrics that describe aspects of biological processes such as wound healing.

We show how Co5M allowed to concurrently study the wound debris removal, re-epithelization, neovascularization, vasodilation, changes in the collagen density, and the plasticity of

sebaceous glands in a single label-free imaging setting and revealed their fine-tuned orchestration across ~7.5 mm² and ~70 days. In particular, we observe that stem cells are activated at early time points and far distances and that their migration is promoted by a degradation of the connective tissue across a large area. We found skin contracture to be initiated first followed by angiogenesis and blood flow increase and that these processes interleave spatially. We further found cleaning and closing of the wound to be in sync with the onset of stem cell activation,

whereas sealing of the former wound site occurs after the vasculature response is completed. In contrast to the so-far presumed healing model of four separated phases and three separated zones, Co5M unraveled a high-order complex entanglement of the tissue responses not observed so far. In the future, Co5M could allow to ascertain disease-caused healing impairments as deviations from the healthy healing process and, subsequently, deduce possible windows of intervention for enhancing the healing effects. Co5M introduces a new paradigm in the applicability of intravital multimodal microscopy for the investigation of spatio-temporal patterns of biological processes involving multiple moieties in translational research, medical diagnostics, and theranostics.

## Results

Co5M (Fig. 1) merges two optical laser excitations, a 532-nm Nd:YAG laser emitting 1.4 ns pulses at 50 kHz for optoacoustic (OA) imaging and a 1043-nm Yb:YAG laser emitting 170 fs pulses at 84.4 MHz for multiphoton (MP) imaging. The two beams are combined by dichroic mirrors and scanned using galvanometric mirrors over an interrogation area (Fig. 1b). The lateral offset between the two beam foci is minimized to ~1 µm by incorporating precisely aligned achromatic doublet lenses and a multinotch dichroic mirror. The axial chromatic aberration between the two foci is ~30 µm in depth, which allows for positioning the OA focus for imaging vasculature in the dermis/hypodermis and the MP focus in the epidermis for connective tissue and sebaceous glands. The SNRs for achieving simultaneous optical excitation with energies below the ANSI limits were determined to be ~34 dB for OA, ~36 dB for 2PEF, ~45 dB for SHG, and ~27 dB for THG. Detection is based on two primary paths. SHG, 2PEF, and BF imaging are done in a reflectance path, through the objective lens using a 0.45 NA 10x objective. Using a multinotch dichroic mirror, the collected backward signals are split to SHG and 2PEF paths and are separated to be detected by highly sensitive photomultiplier tubes while concurrently directing the OA excitation wavelength to the specimen (see Supplementary Fig. 1). THG readings are collected in forward direction through a multimode fiber that is connected to another photomultiplier tube. Ultrasound detection is accomplished in a forward path, i.e., energy that propagates through the sample imaged, using a 50-MHz single-element, spherical-focused ultrasound transducer. The transducer comprises a through-hole to incorporate the multimode fiber for THG sensing and is coupled to the specimen imaged through a water droplet. Co5M achieved imaging resolutions of ~1.1 µm laterally and ~5.7 µm axially for the SHG, 2PEF, and THG mode, and ~1.5 µm laterally for BF (see Supplementary Figs. 2 and 3). OA resolution was ~0.6 µm laterally and ~6.3 µm axially, owing to a diffraction limited optical excitation and a highest detected frequency of 110 MHz. The spatial resolution of Co5M is compromised when imaging biological tissue in vivo due to light scattering as well as microscopic movements of the living tissue to within ~2.8 µm determined by Fourier Shell Correlation (see Supplementary Fig. 8)[64–66], which allows Co5M to image biological microstructures below 8 µm in diameter such as microcapillaries (typically ranging below <10 µm[38,67–69]) and collagen fibrils (typically ranging below <20 µm[29,70–72]) (see Supplementary Fig. 7). An imaging depth of >300 µm is achieved for all multiphoton and optoacoustic modes allowing to appropriately position the imaging planes within the respective tissue layers containing vasculature, connective tissue, and sebaceous glands (see Methods)[11,67,73,74].

Concurrent multimodal acquisition was achieved by a protocol based on a global trigger (Fig. 1c) that electrically synchronizes two data acquisition cards (DAQ1 and DAQ2) and allows for simultaneous acquisition of 4 modes (OA, 2PEF, SHG, THG) with minimized cross-talk using a time-locked acquisition. Streaming data recording based on temporary memory buffers enables continuous multimodal signal acquisition, which is only time limited by the repetition rate of the OA laser, i.e., 50 kHz. Therefore, Co5M allows for a 4-modal signal acquisition at 7 min per square millimeter. Co5M can image large FOVs by translating the specimen continuously in one direction using sample holding stages and raster scanning the two optical foci line-wise back and forth using the galvanometric mirrors to generate a stripe-wise scanning of arbitrary length and ~630 µm maximum width (see Supplementary Fig. 4). Typical FOV in the studies herein were >2.5 × 3 mm², thereby exceeding standard microscopic FOVs.

Applied to wound healing, Co5M imaged mouse ears punctured by a 200 µm-needle over a period of 68 days. We aimed to investigate spatio-temporal dynamics of wound healing hallmarks in a large FOV during the stages of wound healing, revealing microcapillary angiogenesis, collagen enrichment during skin contracture, progression of the wound closure, and plasticity of sebaceous glands as depots for stem cells.

OA, SHG, 2PEF, and THG data (Fig. 2) were acquired at 9 time points during the entire wound healing process for each mouse. Figure 2a–j.1 depicts the microvasculature imaged by OA with vessels ranging from approximately 10 to 200 µm in diameter. The microvasculature network shows fragments of former vessels immediately after the wound infliction, which degrade until day 4, followed by an overall increase in the vasculature density throughout the FOV, which also includes reorganization of the existing network. At day 14, a strong neovascularization of coiled microvessels can be observed close to the former wound region (see Fig. 2j.1), which become normalized by day 36. Figure 2a–j.2 shows the extracellular collagen matrix imaged with SHG. The collagen matrix surrounding the wound initially appears weakened through day 4. A build-up of collagen density after 7 days is seen in the SHG image in Fig. 2j.2. The collagen matrix between the wound and the ear boundary further appears attenuated through day 21. The organization of the collagen as patches structured around hair follicles can be seen in the detailed images. Figure 2a–j.3 plots the 2PEF readings of the tissue autofluorescence, which mainly emanates from elastin and hair follicles, and, thus, reveals the overall tissue appearance. As seen in the 2PEF image in Fig. 2j.3, the wound boundary appears roughly 1 h after the wound infliction and becomes smoothed and homogenized by days 4 and 7. As expected, the skin closes the inflicted wound within 14 days. Figure 2a–j.4 show the THG images of intra-tissue lipid accumulations predominantly in sebaceous glands (see Fig. 2j.4). Despite intensity fluctuations induced by slight intra-measurement focal shifts, the distribution of sebaceous glands appears homogenous throughout the entire mouse ear.

The separate depiction of Co5M's modalities in Fig. 2 visualizes spatial and temporal responses of each imaged biological tissue. The individual observations illustrate specific aspects of the healing process and, thus, reflect and match individually performed experiments by other single-mode microscopes, yet of much higher and equalized image quality regarding resolution and FOV. Whereas separately performed single-mode experiments impede merging of their findings due to differences in the measurement protocol, used animal model, and wound characteristics, joining Co5M's concurrently acquired images evince a distinct underlying coordination in space and time.

The precise spatial registration and fidelity of the individual images depicted in Fig. 2 allows a merged depiction to multimodal images revealing the co-localization of the signals observed. Figure 3 depicts merged hybrid microscopy images that show complex tissue responses during wound healing (see

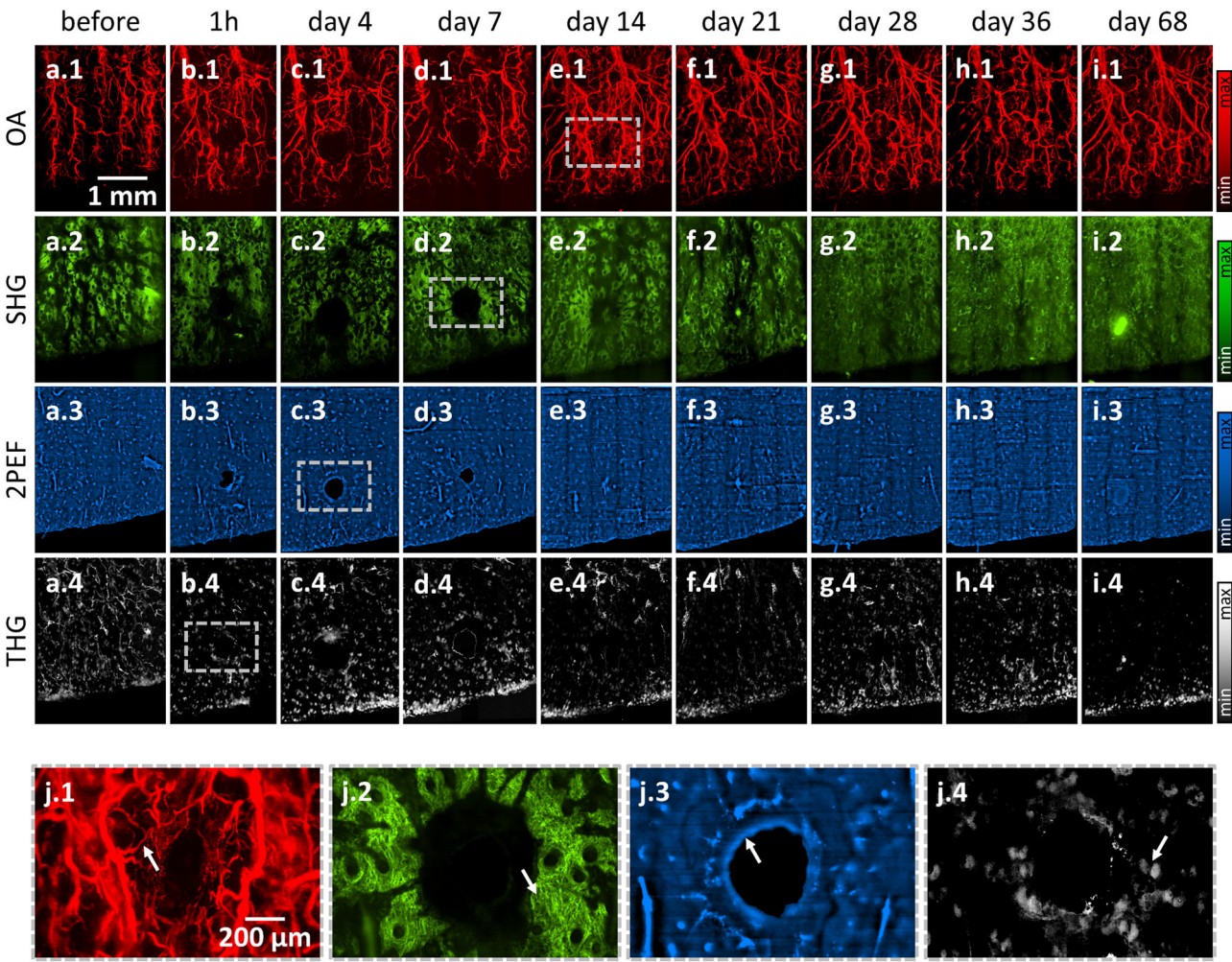

**Fig. 2 Images of a healing wound separated by microscopy modality. a.1** OA, **a.2** SHG, **a.3** 2PEF, and **a.4** THG before wound infliction. Analogous depiction at **b.1–4** 1 h, **c.1–4** 4 days, **d.1–4** 7 days, **e.1–4** 14 days, **f.1–4** 21 days, **g.1–4** 28 days, **h.1–4** 36 days, and **i.1–4** 68 days after wound infliction. **j.1–4** Highlighted zoom-ins showing angiogenesis on the microcapillary level, collagen enrichment, wound boundary smoothing, and visualization of individual sebaceous glands (indicated with arrows).

Supplementary Figs. 5 and 6 for other mice), not revealed by the corresponding brightfield microscopy images (see Fig. 3a–i.2) or photographs (see Fig. 3a–i.3; b.3 is missing to shorten the duration of anesthesia to ensure survival of the mice). The combined Co5M images clearly show that the newly generated rim of tissue at day 4 does not contain collagen, vasculature, or sebaceous glands (see Fig. 3c–e.1) prior to complete wound closure, nor the normal patch-like surface structure of the epidermis. All revealed skin constituents appear recovered and normal at day 68, on which the former wound location is hardly recognizable. The wound healing process can therefore be assumed to have occurred free of complication.

The above results demonstrate the high resolution and temporal precision of the multimodal images obtainable by Co5M. The unique feature of Co5M to acquire its modalities with equalized imaging properties regarding resolution, SNR, and FOV further allows a unified quantitative analysis of the obtained bio-hallmarks. Taking advantage of their precise co-registration in space and time achieves to unite, compare, and relate the hall-mark's responses for understanding of the wound healing process. We were particularly interested in describing metrics to grasp the spatio-temporal interplay of angiogenesis, blood supply, skin contracture, and involvement of stem cell depots. Figure 4a–c quantifies morphological changes of the wound and its boundary.

We can see that the wound initially increases in size (Fig. 4a), becomes rounder (Fig. 4b), and smoother (Fig. 4c), prior to all three metrics reversing during wound closure, possibly indicating a reorganization of the wound boundary before re-epithelization begins. Figure 4d–g depicts quantifiable changes in specific hall-marks of wound healing as functions of distance to the wound and time after wound infliction. Figure 4d quantifies the branching of the vasculature network using a fractality score normalized to the image obtained before injury. We observe the appearance and disappearance of the injury in the tissue (blue area, fractality score <0.8), as well as an increase in fractality between days 4 and 14 (yellow-red area, distance to wound ca. 300–400 μm). In fact, the fractal dimension of the vessel network in the mouse ear has been approximated from optical micro-angiography images[35], finding that $\dim_{box}(V) \approx 1.75$. Our investigation (more or less) confirms this finding, with a median fractal dimension of the mouse ear vessel network of 1.71, locally reaching values up to 1.83 during neovascularization. Figure 4e shows a visualization of the change in median vessel thickness normalized to the before-image. We observe a strong increase in vessel size after day 7 at a distance of approximately 400–500 μm, as well as the approach of enlarged vessels to the wound center between 7 and 21 days. Figure 4f depicts the relative median collagen density normalized to the before-image. Although a

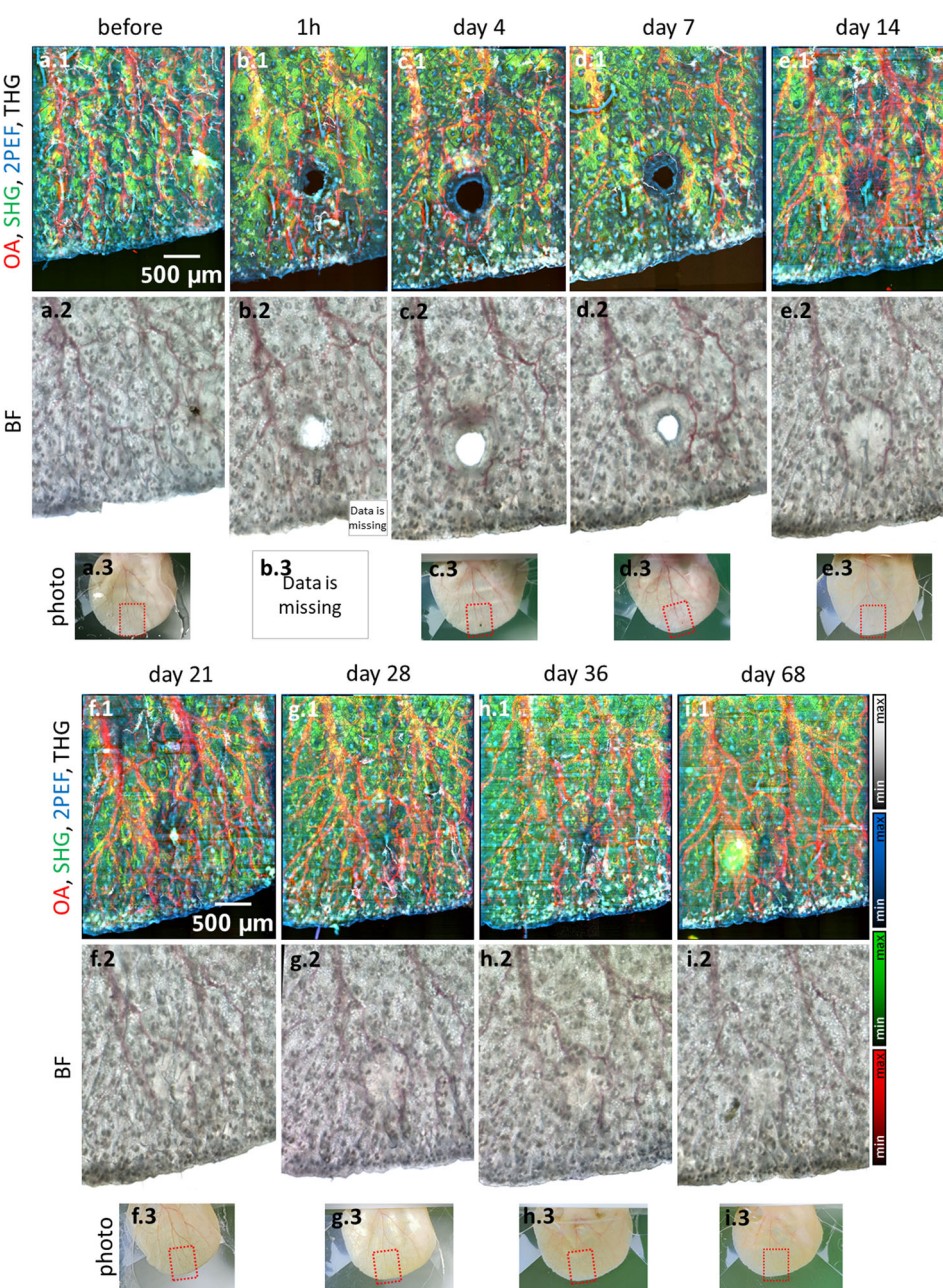

**Fig. 3 Longitudinal imaging of the wound healing process in a mouse ear in vivo by label-free multimodal microscopy achieved using Co5M. a.1** Multimodal image of the mouse ear before wound infliction and corresponding **a.2** brightfield microscopy image as well as **a.3** photograph of the ear. Analogous depiction of the ear at **b.1–3** 1 h, **c.1–3** 4 days, **d.1–3** 7 days, **e.1–3** 14 days, **f.1–3** 21 days, **g.1–3** 28 days, **h.1–3** 36 days, and **i.1–3** 68 days after wound infliction.

strong inter-image variation prevents detailed conclusions, we observe a decrease in collagen density up to a distance of ~450 µm between 1 h and 14 days after wound infliction, as well as a general trend of increasing collagen density at a distance of ~400 µm starting after 7 days and at the former wound site after 28 days. Figure 4g visualizes the movement of sebaceous glands as the relative spatial distance to the wound center compared to the before-image. We found a general trend of sebaceous glands moving towards the wound occurring over a large area up to ~700 µm away from the wound. The glands remained in these positions until day 68. Furthermore, until day 68, we observed in all mice the formation of at least 2 new sebaceous glands in freshly built tissue and at least 2 existing sebaceous glands migrating into the former wound region.

As seen in Fig. 4h–o, the maximum intensities of the spatial and temporal projections of the data can be separately examined allowing for further detailed quantitative analyses of key metrics of the healing process. For each metric (vasculature fractality, vessel thickness, collagen density, and sebaceous gland movement), the maximum intensity projections are plotted for the individual mice (gray lines, mice 1–3; cyan line, average), along with response average (orange line) and standard deviation (orange shade). The location and time of the respective response maxima can be quantified by applying a simple Gaussian model (Fig. 4a–c, h–o, black lines; see Table 1 for extracted values). The numerical data shown in Table 1 allows the fast visualization of relationships and trends.

The characterization of the wound geometry regarding size, shape, and boundary indicated cleaning and closing of the wound

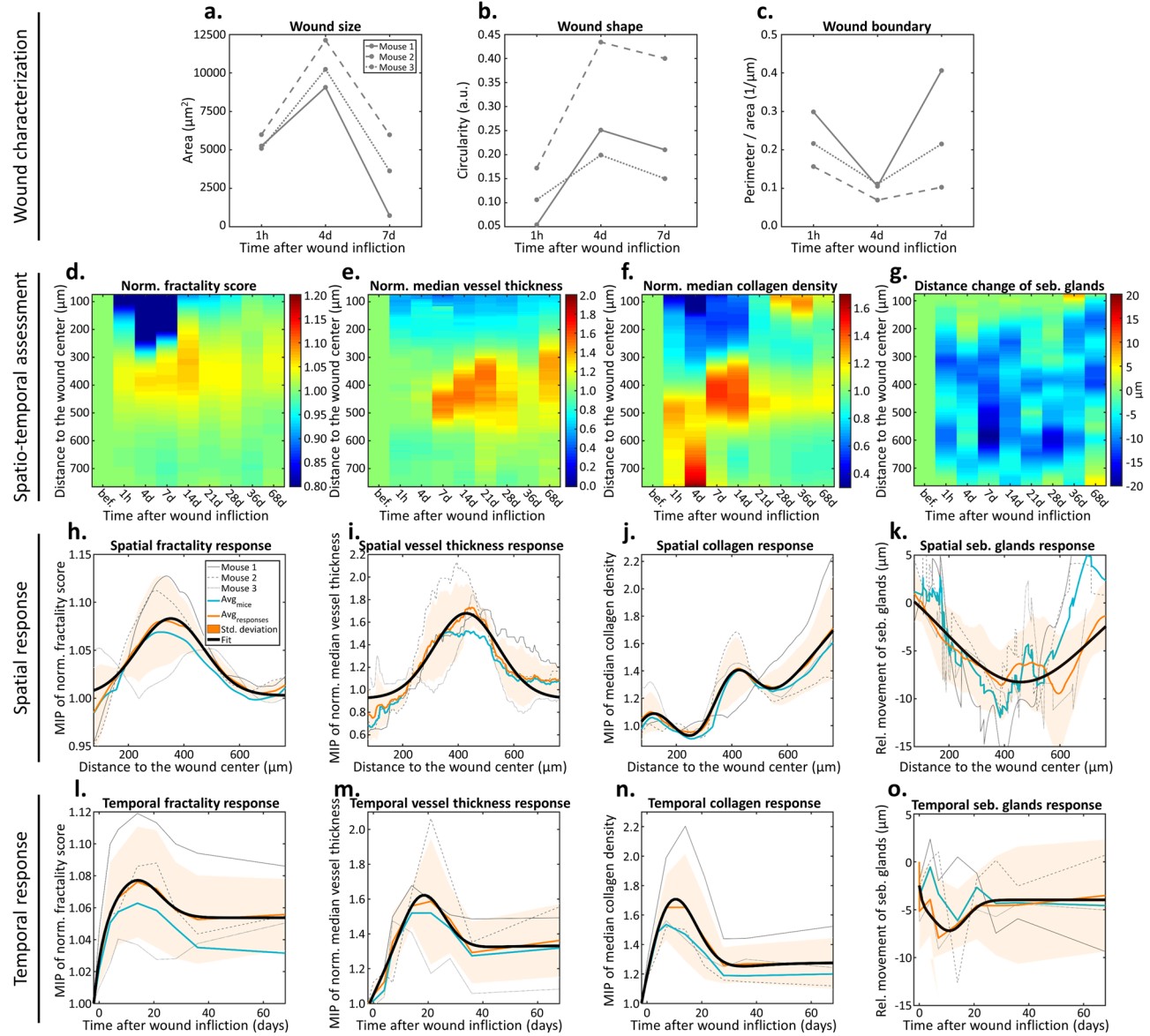

**Fig. 4 Spatio-temporal assessment of the wound healing process via averaging the analysis of OA, SHG, 2PEF, and THG images of three mice.** Spatial characterization of the wound towards **a** the wound size, **b** the wound shape, and **c** the wound boundary. Spatio-temporal maps depicting relative changes of **d** the fractality and **e** vessel thickness of the microvasculature as well as **f** the local collagen density and **g** distance change of sebaceous glands to the wound center compared to the image recorded before wound infliction. Maximum intensity projections (MIP) along the spatial dimension of **h** the normalized fractality score, **i** the normalized median vessel thickness, **j** normalized median collagen density, and **k** relative movement of sebaceous glands. **l–o** Analogous analysis along the temporal dimension.

via debris removal and re-epithelization within the first 5 days after wound infliction. By localizing the increase in vasculature fractality in space, we observed microcapillary angiogenesis occurring at a radius of ~350 µm (roughly double the wound radius) with a width of 280 µm. An analogous analysis of the increase in vessel thickness showed an enhancement in blood flow further away at a radius of 430 µm and with a similar width of 270 µm. We found the collagen density increased at $400 \pm 290$ µm, which is indicative of skin contracture occurring between the location of angiogenesis and increased blood flow. Collagen formation was also observed between 0 and 180 µm from the wound center, corresponding to the sealing of the former wound site. In contrast, collagen density decreases at $240 \pm 240$ µm, suggesting skin loosening adjacent to the former wound perimeter. Tracking the sebaceous glands, we observed an overall tendency of migration towards the wound across almost

the entire FOV. A similar analysis of these hallmarks in time showed skin contracture and loosening occurred earliest together with the migration of sebaceous glands, all of which began at ~8–9 days after injury and continued for ~7 days. We observed angiogenesis starting after these responses at ~14 days and continuing for ~10 days. The increase in blood flow began at roughly the midpoint of angiogenesis (~18 days) and extended over ~9 days. All metrics enter a long-lasting plateau phase, indicating that the associated hallmarks approach, but never reach, their original condition before injury. Despite the healing process considered as completed, a long-lasting difference of the monitored hallmarks when compared to the before-image can be interpreted as very mild scarring[75,76].

The quantitative analyses using simple Gaussian models further allowed a schematic visualization of the associated intratissue processes in space and time (Fig. 5). Figure 5a shows the spatial

**Table 1 Quantitative assessment of the wound healing process via monitoring key metrics in space and time.**

| Key Metric | Dimension | Peak | FWHM | Transition | $R^2$ | Hallmark |
|---|---|---|---|---|---|---|
| Wound size | Area | 1115 $\mu m^2$ | / | / | / | Wound closure via debris removal and re-epithelization |
| | Time | 3.1 days | 6.0 days | / | / | |
| Wound shape | Circularity | 0.31 a.u. | / | / | / | |
| | Time | 4.9 days | 8.0 days | / | / | |
| Wound boundary | Perimeter/Area | 0.92 $\mu m^{-1}$ | / | / | / | |
| | Time | 3.35 days | 6.1 days | / | / | |
| Vasculature fractality | Space | 353.0 $\mu m$ | 280.4 $\mu m$ | / | 0.966 | Angiogenesis |
| | Time | 13.7 days | 9.8 days | 19.9 days | 0.996 | |
| Vessel thickness | Space | 428.9 $\mu m$ | 271.7 $\mu m$ | / | 0.916 | Blood flow increase |
| | Time | 17.6 days | 9.1 days | 21.4 days | 0.969 | |
| Collagen density | Space | 80.0 $\mu m$ | 186.8 $\mu m$ | / | 0.995 | Wound sealing |
| | Time | 30.3 days | 20.9 days | 36.3 days | 0.860 | |
| | Space | 240.7 $\mu m$ | 236.0 $\mu m$ | / | 0.995 | Skin loosening |
| | Time | 16.6 days | 22.4 days | 37.5 days | 0.924 | |
| | Space | 402.1 $\mu m$ | 290.9 $\mu m$ | / | 0.995 | Skin contracture |
| | Time | 8.08 days | 7.1 days | 18.5 days | 0.992 | |
| Movement of seb. glands | Space | 450.8 $\mu m$ | 632.1 $\mu m$ | / | 0.884 | Plasticity of stem cell hubs |
| | Time | 8.9 days | 7.1 days | 17.9 days | 0.685 | |

Relative changes of the key metrics are fitted using Gaussian models to localize their response regarding the spatio-temporal location (peak), the spatio-temporal extent (FWHM), and potential timepoints marking the transition to long-lasting plateau phases.

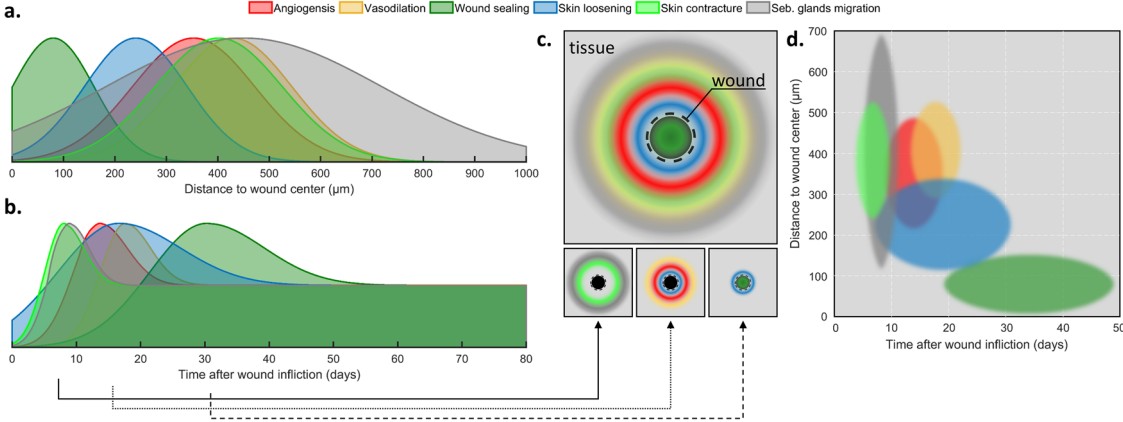

**Fig. 5 Wound healing model in space and time based on spatio-temporal assessment by Co5M. a** Temporal sequence and **b** spatial coordination of the investigated hallmarks. **c** Schematic depiction of the healing process in zones surrounding the wound, highlighted for three time points at day 8, 16, and 31. **d** Spatio-temporal diagram of healing hallmarks.

location and spread across which the analyzed hallmarks of wound sealing occur. Figure 5b reveals a distinct temporal sequence of these hallmarks and their transition to a long-lasting phase. In both cases (spatial and temporal), a coordinated scheme of the hallmarks can be observed, which also overlap to a non-negligible amount. Figure 5c schematically depicts the respective rims of the hallmarks surrounding the wound, whereas the three selected timepoints at day 8, 16, and 31, and the spatio-temporal diagram in Fig. 5d illustrate the overall healing process to start at more distant locations and proceeds towards the wound center.

## Discussion
Biological tissue is composed of micrometer-sized functional units, including microcapillaries, collagen fibrils, skin and stem cells, and skin appendages, among others, that maintain and reconstitute the tissue's integrity and function in a coordinated and interdependent sequence of functionalities. Therefore, understanding complex biological processes requires concurrent monitoring of the process' essential functional units across the entire affected area, without disturbing the tissue, as well as the investigation of the process' coordination in space and time. It is

particularly important to elucidate the entanglement of the tissue layers, functional units, and skin appendages involved in the investigated biological process and to resolve their mutual interdependency[1,7,9,14,18]. Such comprehensive information can only be obtained by an imaging system that can access multiple biological contrasts to formulate a schematic model of the process that forms the basis to deduce disease-caused impairments and, subsequently, the most suitable treatment options.

To this end, we developed a concurrent 5-modal microscope (Co5M) that allows congruent and concurrent investigation of multiple functional biological moieties. The underlying concept is that the combination of multiple microscopy modes can capture biological mechanisms to a level of detail and complexity not accessible by single-mode microscopes. However, previous implementations required that each mode was acquired sequentially, which reduced spatial and temporal co-registration, and large observations fields were only accessible via manual stitching of the ~(0.6 mm)$^2$ FOV. Whereas this performance reflected standard imaging capabilities of intravital microscopy, it did not allow the study of biological processes characterized by micrometer-sized functional units, which longitudinally interact in a highly complex entanglement across millimeter-sized areas.

To this end, we augmented the working principle of Co5M over previous installments to now feature micrometer precision, millimeter-sized FOVs, and label-free interrogation within pre-clinical time-frames. Highly co-registered performance of the equipped modalities in space and time was achieved by implementing fully automated triggering and scanning schemes, acquiring data using high-speed streaming, and optimizing the optical assembly. The multimodal contrast allows Co5M to resolve the key characteristics of biological tissue, such as vasculature, connective tissues, skin appendages, and morphology, and can superimpose these findings onto the tissue's anatomy.

We applied Co5M to study a healing wound, a task that challenges the limits of microscopy systems by demanding simultaneous tracking of cascading responses of multiple tissue components to the injury over an extended area and duration. The large FOV and precisely co-registered modalities of Co5M afforded the most information-dense images of all stages of the wound healing process to date, allowing the high-resolution overlay of the essential microscale functional units, such as microvessels (OA), collagen fibrils (SHG), and sebaceous glands (THG), as well as skin morphology (2PEF). Previous microscopy studies have provided scattered insights into individual healing mechanisms or biomarkers on a multitude of animal and wound models under a variety of conditions, mostly focusing on only one or two functional units, imaging only a small FOV, selecting too few time points, or lacking in the extraction of a schematic and comprehensible wound healing model. Co5M addresses the core limitations of these works and overcomes the difficulty of merging their outcomes by yielding comprehensive, high-fidelity data on all relevant biomarkers in a single imaging setup.

We further aimed for the quantitative extraction of these processes' dynamics to generate a wound healing model that encompasses the interdependent series of healing events over time and space. The high fidelity of the data provided by Co5M enabled a comprehensive quantification of the spatial and temporal responses of tissue components to an injury allowed us to localize the temporal and spatial activations of the monitored bio-hallmarks by fitting Gaussian functions to the data. We defined five metrics (wound geometry, vasculature fractality, vessel thickness, collagen density, and movement of sebaceous glands) to assess seven specific hallmarks of the healing process (wound closure, angiogenesis, vasodilation, wound sealing, skin loosening, skin contracture, and plasticity of stem cell depots). The congruent monitoring of these hallmarks allowed Co5M to localize the different healing events surrounding the wound (Fig. 5a), while the simultaneous monitoring allowed identification of their onsets, peaks, and durations (Fig. 5b). Although separate quantifications of previous studies align with our data (though with poorer fidelity) regarding irregular wound closure[3,11,36], avascular re-epithelization[3,11,36], microcapillary angiogenesis[11,22,34,36,38], vasodilation[3,34,36], skin contracture[11], wound sealing[11,15,32], or stem cell activation[11,18,19], Co5M allows the formulation of a far more accurate scheme of the spatial and temporal orchestration of these hallmarks than previously possible (Fig. 5c, d).

Co5M provides higher precision and simultaneous access to more biological components, facilitating more detailed descriptions of healing events and insights into previously unobserved interdependency of the bio-hallmarks. For instance, we observed two responses of the vasculature (i.e., angiogenesis and vasodilation) interleaving spatially and temporally with three responses of the connective tissue (i.e., wound sealing, skin contracture, skin loosening). We also found the highest activity of the overall healing process to begin at more distant locations and proceed over time towards the wound (Fig. 5c, d). Such new insights constitute the basis for a more detailed understanding of non-optimal healing (e.g., infection, scarring, chronic wounds, diabetic

ulcers), including which hallmarks are affected, prolonged, and dysregulated, and subsequently what treatments may be most suitable[20,21]. Co5M can, in the future, shed light onto unresolved interdependent processes and associate them with underlying conditions, such as diabetes, sepsis, or infections.

Co5M's combination of simultaneous, multimodal, and label-free capabilities, which are applicable across a large FOV with sub-cellular precision and high SNR, makes it a versatile platform for longitudinal studies of nearly any process in accessible tissue that is characterized by the key biological functional units discussed here. Whereas the current configuration of Co5M allows a maximum sample thickness of <500 μm due to its transmission mode operation, first steps were taken towards redesigning the interrogation area for epi-mode imaging[49,77–79], facilitating the investigation of thicker specimen. To counteract the inherent maximum imaging depth of ~300 μm for the optical resolution microscopy modalities used herein, multiscale imaging methods and approaches, especially for the optoacoustic mode, would allow to fuse microscopic with mesoscopic findings[52]. Furthermore, whereas the chromatic aberration of the applied excitation wavelengths was beneficial to image various biological moieties at different depths within the skin, implementing reflective focusing elements could allow Co5M to apply multimodal microscopy on a single-cell level without refocusing or post-processing. The optoacoustic mode of Co5M complements the optical readings by yielding absorption measurements that will in the future provide label-free access to biomarkers beyond blood, such as lipids, carbohydrates, proteins, and water[80–82]. Such capabilities and the option to make use of specific molecular reporters for Co5M's modalities[50,83–86] could allow the system to visualize an even larger number of contrasts and target crucial biological constituents like macrophages, lymphatic vessels, and cellular metabolites[1,7,11,19–21,32]. Co5M paves the way for a new kind of transformative biomicroscopy, enabling new insights into life on the microscale with so far unmet precision and completeness.

## Methods

**Optical setup.** The two subsystems of Co5M, namely the multiphoton part and the optoacoustic part, are merged into a single optical setup. The combined system, as schematically depicted in Fig. 1, integrates four microscopy modalities by ultra-sensitive microscopy photomultiplier tubes (PMT) in a way in which that they can operate spatially and temporally aligned, meaning at the same time and at the same position. Co5M is a laser-scanning microscope based on a set of high-precision galvanometric (galvo) mirrors (6215H Galvanometer Scanners, Cambridge Technologies) in combination with an inverted microscope stand (AxioObserver.D1, Zeiss) as the shared central scanning unit of the microscope. Subsequent to a beam expansion using achromatic doublet lenses (AC254-050-B-ML and AC254-300-B-ML, Thorlabs) in a 4f-configuration, the excitation beams are coupled into the inverted microscopic stand and are focused by infinity corrected microscopic objective lenses (Plan-Apochromat 10x/0.45). In order to provide best optical focusing and automated system control, a high-precision endless rotational stage (DT-34, Physik Instrumente) is attached to the fine drive of the microscope stand. The sample is positioned by set of motorized xy-stages (MLS203-2, Thorlabs) together with a piezoelectric z-stage (MZS500-E, Thorlabs) adapted to the microscope stand. The sensors positioned in transmission-mode, i.e., above the sample, was attached to another set of high-precision motorized xyz-stages (M-683.2, M-501.1, Physik Instrumente).

The multiphoton microscopy (MPM) subsystem equips an Yb-based solid state femtosecond laser emitting near-infrared light at a wavelength of 1043 nm as the excitation source (YBIX, Time-Bandwidth; pulse width: 170 fs, average power: 2.8 W, repetition frequency: 84.4 MHz) as schematically depicted in Fig. 1. All multiphoton signals are detected by ultra-sensitive photomultiplier tubes (PMT) (H9305-03, Hamamatsu), which are further amplified (C7319, Hamamatsu) and gain-controlled by a linear DC voltage output supply (GPD-4303S, GW Instek). Two of the PMTs are installed in an epi-configuration in combination with a motorized filter wheel (FW102C, Thorlabs) for sensing SHG, and 2PEF, and another PMT in transmission-configuration for sensing THG via a multimode fiber. In order to assign the PMTs specifically to these multiphoton modalities, appropriate optical filters are installed (SHG: FB520/10, Thorlabs; THG: FGUV11, Thorlabs; 2PEF: FGS900, Thorlabs; see Supplementary Fig. 1). Further, a longpass dichroic mirror (DMLP550, Thorlabs) is separating SHG and 2PEF signals in epi-configuration. For simultaneous optoacoustic microscopy at 532 nm, each PMT is additionally equipped with notch filters (NF533-17, Thorlabs) to prevent crosstalk and potential damage of the optical sensor by the optoacoustic laser.

The optical-resolution optoacoustic microscopy (OAM) subsystem operating in time-domain uses a diode-pumped solid-state laser (SPOT-10- 200-532, Elforlight Ltd; pulse width: 1.4 ns, actively Q- switched, average power: 2 W, repetition rate: up to 50 kHz, emitted wavelength: 532 nm). For sensing the generated optoacoustic signals, a spherically focused single-element ultrasound transducer (HFM31, Sonaxis; central frequency: 53.4 MHz, -6 dB bandwidth: 29.5–77.3 MHz (90%), focal distance: 4 mm, central hole: 1 mm diameter) was positioned in transmission-mode by the previously mentioned xyz-stages. The ultrasound transducers achieved acoustic focusing by a concave glass lens attached to the active element constituting the actual acoustic sensor. The HFM31 transducer comprises a central hollow metal sleeve for THG-fiber guidance. The transducer was acoustically coupled to the samples by a combination of centrifuged ultrasound gel squeezed onto the sample by a thin polyethylene foil and a droplet of deionized water to provide best acoustic transmission from the sample to the sensor. In order to match the sensitivity field with the entire scanned area, the used transducer is positioned in positive defocus. The detected signals were further amplified by a low-noise 63 dB amplifier (AU-1291, Miteq) and digitized by a high-speed 12-bit DAQ card (ADQ412, SP Devices; max. sampling rate: 3.6 GS/s) in a streaming-like acquisition mode.

**Data acquisition**. Synchronization and simultaneous data acquisition (see Fig. 1) of Co5M is facilitated using two data acquisition cards, hereinafter called DAQ1 and DAQ2 (DAQ1: PCIe 6363, National Instruments; max. sampling rate: 1 MS/s; and DAQ2: ADQ412, SP Devices; max. sampling rate: 3.6 GS/s). The entire multimodal scanning procedure is programmed in a self-developed MATLAB$^{TM}$ script (see Appendices: MATLAB$^{TM}$ scripts). DAQ1, which is connected to a BNC connector block (BNC-2110, National Instruments; 16 analog input/output connections, 2 digital input/output connections, range: ±10 V), sends out a trigger signal; hereafter called global trigger signal. This global trigger signal determines the scanning frequency, the acquisition frequency, as well as the repetition rate of actively triggered lasers for optoacoustic microscopy and constitutes a 0–5 V TTL signal as a digital waveform. DAQ1 further sends out two analog waveforms ranging across ±2 V as the slow-axis and fast-axis scanning curves for the galvo mirrors and digitizes the multiphoton signals as detected by the three PMTs on a ±10 V range. These analog signals are binned according to the underlying control frequency. As the optoacoustic laser is actively triggered by the sent-out On-states of the global trigger signal, only the Off-states are used for recording the multiphoton signals to prevent cross-talk or sensing optical signals excited by the optoacoustic laser. DAQ2 is used for reading the high-speed data such as the optoacoustic signals. For synchronization purposes, the global trigger signal provided by DAQ1 is also fed into the trigger input of DAQ2. As DAQ2 runs at a very high sampling frequency and accepts incoming signals in the range of ±750 mV, electrical attenuators and a high-pass filter (EF515, Thorlabs) are installed to shape the incoming TTL signal of 50% duty cycle appropriately to a sharp trigger peak. Hence, both data acquisition cards are electrically synchronized. Based on ~200 MHz being the highest frequency expected to be sensed by the equipped ultrasound transducers, the sampling rate of DAQ2 is set to 450 MS/s in order to fulfill the Nyquist–Shannon sampling theorem. Further, a streaming-like acquisition mode was developed being only restricted by the data transfer rate of the PCIe-connection of DAQ2 to the computer (i.e., 3.2 GB/s) and the memory capacity of the used computer (i.e., 64 GB in case of the used PC). The streaming-like acquisition mode is based on assigning temporary data buffers internally on the DAQ2 memory, in which the recorded data can be continuously filled. This approach is therefore called Continuous Multirecord (CMR). The basic principle of CMR is the definition of temporary buffers which are readout during data recording. In detail, the internal storage of DAQ2 is subdivided into shorter data buffers of data point length 4096. By starting the data acquisition, the first buffer will be filled with signals until the internal loop coded in the acquisition algorithm returns to start. The duration of the first loop of records depends on the underlying scanning frequency, the record length per data point, as well as background operations of the computer itself. As the latter vary over time, the number of data points per loop is not constant. The size of the temporary buffers is hence selected to be sufficiently large. The first loop of records received e.g., $x$ data points which are stored in the first $x$ entries of the first temporary buffer. As the data recording continues, the second loop assigns the recorded data to the second buffer, and so on and so forth. Meanwhile, in order to record the remaining data in such a continuous manner, the first buffer partly filled with data is fetched to the computer memory, freed after complete data transfer, and is queued to the end of the buffer chain again. In this way, the data is transferred simultaneously to the recording, requires minimal extra time after recording the last data points, constitutes the fastest acquisition mode of large data sets, and is not limited by the internal storage capacity of DAQ2.

**Image scanning procedure**. The image formation and scanning procedure takes advantage of the synchronized and simultaneous acquisition of all modalities equipped in the Co5M system as described. Co5M offers high spatial controllability and automation of all relevant parts required for the image scanning: movement of the optical focus in $x$, $y$, and $z$; movement of the sample in $x$, $y$, and $z$; and movement of the detector in $x$, $y$, and $z$. Co5M takes advantage of the fully automated control to combine the above mentioned scanning procedures in a

hereinafter called hybrid-scanning mode. In this scanning procedure, the sample holding stage is moving in one direction continuously, whereas the galvo mirrors are scanning the optical foci along a straight line back and forth perpendicularly to the sample movement (see Supplementary Fig. 4). Hybrid-scanning enables the acquisition of stripes of 638 μm maximum width and nearly unlimited length with maximum spatial precision. Whereas the OAM recordings are intrinsically capturing a depth-profile along the z-axis, no optical z-stack imaging was performed for the MPM modalities. The combination of the fast bi-directional scanning of the galvos with a continuous translation of the sample represents the fastest and most elaborated scanning procedure for ROIs larger than the maximum FOV achieved by laser-scanning only. The imaging protocol consisted of (1) anesthetizing the mouse and waiting for the mouse's physiological conditions to stabilize (~2 min); (2) positioning and preparing the ear for imaging (see Methods—Animal Model) (~2 min); (3) aligning the FOV and adjusting the optical focusing (~1 min); (4) acquiring the simultaneous multimodal imaging (~50–60 min); (5) acquiring the brightfield microscope images (~5 min); (6) capturing a photo of the ear (~0.5 min); disassembling the holder (~2 min); and (7) watching the mouse recover from the anesthesia (>5 min). The total duration of anesthesia was therefore below 1 h 20 min. For the imaging 1 h post wound infliction (see Fig. 3b.1–3, Supplementary Fig. 5b.1–3, and Supplementary Fig. 6b.1–3), the anesthesia was reduced to 1 h by capturing a slightly smaller FOV and by omitting the photograph to ensure survival of the mouse.

**Image analysis**. The imaging was carried out using the hybrid-scanning scheme covering a FOV of ~2.5 · 3 mm² located at the outside edge of the pinna. Furthermore, all modalities are recorded simultaneously, and were complemented with brightfield microscopy images of the identical FOV. The recorded data of OA, SHG, 2PEF, and THG images were co-registered to each corresponding BF image. The high precision data and large scanned area allowed to apply spatio-temporal analysis techniques to explore the tissue response as a function of time and of distance to the wound. To ensure comparability and spatial equality, the sets of images of each mouse were co-registered along the time course.

The OA images were analyzed to monitor changes in the microvasculature. For that purpose, OA images were first processed with an anisotropic diffusion filter in 10 iterations to reduce image noise. Next, tubularity filters were applied on the entire images as well as on moving windows to generate binary masks, which were finally merged to receive a segmented vasculature map. These maps were subsequently analyzed in terms of different local features. On the one hand, the median vessel thickness was determined by measuring the local vessel thickness using a central circle analysis. On the other hand, the morphology of the vasculature tree has been analyzed via a fractality measure. The fractality of a vessel network quantifies the hierarchical complexity of the network in terms of its ability to fill an area (or a volume) with its branches. More precisely, we used the box counting dimension (also called Minkowski–Bouligand dimension) $\dim_{box}(V)$ of a vessel network $V$, that is defined by

$$\dim_{box}(V) := \lim_{n \to \infty} \frac{\log N(1/n)}{\log n}, \qquad (1)$$

where $N(1/n)$ is the number of boxes of side length $1/n$ needed to cover the vessel network $V$. Numerically, this quantity is locally approximated in patches of $128 \cdot 128$ μm². The box counting dimension of a network is a dimension in the sense that it gives a scaling law for the area occupied by the set: $N(1/n) \sim n^{\dim_{box}(V)}$. For a set $V$ that is a line or a surface, the box counting dimension agrees with the classic integer dimension and equals one and two, respectively. Since dense vessel networks fill an area quite efficiently, their box counting dimension is expected to be close to two.

The SHG images mapping the extracellular collagen matrix were analyzed to reveal the skin contracture induced by the connective tissue. In here, the SHG images were normalized within itself to compensate for inter-image fluctuations and were analyzed towards the median value of local signal intensity. Further, the extracted median intensities were normalized across the analyzed region to yield a local collagen density. Spatio-temporal analysis maps of OA and SHG signals were further normalized to the before-image to depict relative changes.

**Monitoring tissue hallmarks**. The 2PEF images were utilized to monitor the wound size and boundary condition. For that, the 2PEF images were normalized to itself, the wound was outlined using a tracing tool with a tolerance of 5% in legacy mode, and the traced area as characterized by its area, perimeter, and circularity. The above mentioned spatio-temporal maps of each mouse as well as their average were projected along the spatial or temporal dimensions using their maximum intensity. The averaged curves were subsequently fitted by Gaussian functions. More precisely, we used the model

$$m(s) = a \cdot e^{-\frac{(s-b)^2}{2c^2}} + d \cdot s + f \qquad (2)$$

for the spatial responses using $s$ as the spatial distance to the wound center (the spatial collagen responses were fitted by a sum of two Gaussian functions with a

linear slope of $d \neq 0$; all other fits had a static offset with $d = 0$), and the model

$$m(t) = a \cdot e^{-\frac{(t-b)^2}{2c^2}} + d \cdot \left(g - e^{-(t-h)}\right) + f \quad (3)$$

for the temporal responses using $t$ to denote the time after wound infliction.

**Animal model**. All animal procedures were approved by the Government of Upper Bavaria. The wound healing process was studied using three athymic nude Hsd Foxn1 mice (sex: female; age: 6 weeks) and the wounds were inflicted by punching the ear with a 200 µm sterile needle with a trimmed bevel in order to stamp out a piece of tissue. The mice were imaged consecutively up to 68 days after infliction of the wound according to typical settings performed in analogous studies. An automated physiological monitoring system is installed for mouse in vivo measurements (PhysioSuite, Kent Scientific). This system monitors heart- and breathing-rate, blood oxygenation, and body temperature of the mice. The latter is controlled in a fully automated feedback scheme using heating foils (thermo polyester heating foils, Conrad) being placed beneath the mice. Furthermore, a small animal anesthesia unit (Tabletop Veterinary Anesthesia Machine, Keebovet) is used for mouse in vivo measurements. During measurements, the mice were anesthetized with ~2% isoflurane and placed onto a dedicated mouse holder. The end pieces of both, the monitoring and the anesthesia system for mouse in vivo measurements, are fixed to a custom-designed mouse holder fitting on the sample holding stages. This holder contains an elevated pedestal comprising a central hole across which a 170 µm cover slip is placed. On top of the cover slip, the mouse ear is placed and fixated. A drop of ultrasound gel can be squished on top of the ear with a thin polyethylene foil wrapped around a hollow frame, which is finally screwed to the mentioned pedestal. In this condition, the mouse ear is sandwiched between the glass cover slip from below and the ultrasound-gel-pressing foil from above. By that, the ear can be accessed optically from below by elevating the equipped microscopic objective lens and acoustically from above by lowering the ultrasound transducer into the hollow frame filled with water. The holder has been iteratively optimized using a 3D printer (RF1000, Renkforce) and finally fabricated of aluminum to enhance stability.

**Wound healing model**. Mammalian wound healing is a complex process in both spatial and temporal terms, presenting challenges for possible examination techniques. On the one hand, healing wounds are based on a finely tuned and coordinated response of multiple tissue layers of the skin involving the epidermis, the dermis, and the hypodermis[1,9,14] and associated cellular components such as basal and differentiated keratinocytes, collagen and elastin, fibroblasts, adipocytes, and other components of the extracellular matrix[1,9,14]. The thickness of these tissue layers typically ranges about ~100 µm for the epidermis containing the connective tissue and sebaceous glands, ~100 µm for the dermis/hypodermis containing shallower microvasculature and deeper bigger vasculature structures[11,67,73,74]. The area affected by wounds can be coarsely subdivided as a function of distance to the wound center in zones of former wound site (depending on the inflicted wound; typically for mouse models ~0.5–2 mm), the leading edge (typically up to 0.5 mm away from the wound boundary), and the proliferative hub (0.5–1.5 mm away from the wound boundary)[3,11,18,19]. The wound site typically first increases by debris removal and then decreases, mostly in an irregular and asymmetric shape due to skin contracture. The leading edge proceeds to close the wound in form of an initial avascular region, which was sealed by an increase in connective tissue and filled by freshly formed microcapillaries after wound closure. The proliferative hub is characterized by activation of stem and epithelial cells, regeneration of sebaceous glands, and increase in blood supply. On the other hand, the wound healing process typically follows a sophisticated sequence divided in four essential phases[1,3,7–17]: In the first phase (hemostasis), blood thickens and clots in order to seal the wound and prevent excessive blood loss. During the second phase (inflammation), macrophages and white blood cells absorb bacteria and debris in the wound via phagocytosis. Simultaneously, the extracellular collagen matrix gets degraded in close proximity to the wound by enzymes to facilitate migration of epithelial cells. The third phase (proliferation) starts after closure of the wound by newly formed skin (epithelium). In this phase, new vasculature is built by angiogenesis, the connective tissue is reformed by collagen and elastin deposition, and new skin is generated by granulation tissue formation and epithelialization. This phase is further characterized by skin contraction. In the final phase (remodeling), newly built tissue is refined to standard conditions by rearrangement of the connective tissue, intramolecular crosslinking, and reorganization of the vasculature.[1]

**Reporting summary**. Further information on research design is available in the Nature Research Reporting Summary linked to this article.

## Data availability

Source data for the charts in the main figures is available as Supplementary Data 1 and any remaining information can be obtained from the corresponding author upon reasonable request.

## Code availability

The code that supports the findings of this study is available from the corresponding author upon reasonable request.

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

## Acknowledgements

The authors thank Pia Anzenhofer for her assistance in mouse preparation, wound infliction, and in vivo measurements, as well as Robert J. Wilson for his support in drafting this manuscript.

## Author contributions

M.S. conceived the idea of augmenting the working principle of concurrent multimodal microscopy to image healing wounds. M.S. conceptualized and built the optical system, the data acquisition and triggering scheme, and implemented the control code for the microscope. M.S. characterized the system imaging capabilities, performed all imaging experiments of the healing wounds, and processed the images. C.D. and D.J. conceived the idea of quantitatively analyzing the images and to extract metrics capturing the bio-hallmarks. C.D. carried out the image analysis. M.S. and V.N. wrote the manuscript and all authors read and edited the paper.

## Funding

This work was supported by the Deutsche Forschungsgemeinschaft (DFG) as part of the CRC 1123 (Z1), the DFG Reinhart Koselleck project (NT 3/9-1), and the DFG Gottfried Wilhelm Leibniz Prize (NT 3/10-1). Open Access funding enabled and organized by Projekt DEAL.

## Competing interests

The authors declare no competing interests.
