## [Peer Review File · Communications Biology]

Reviewers' comments:

Reviewer #1 (Remarks to the Author):

This manuscript, "Label-free concurrent 5-modal microscopy (co5M) resolves unknown spatio-temporal processes in wound healing", presents a new concept of the co5M including optoacoustic, two-photon excitation fluorescence, second harmonic generation, third harmonic generation and bright field microscopy. The co5M can simultaneously monitor various metrics of the wound healing process in the mouse ear. The authors systematically summarized the imaging mechanism and well explained the role and performance of each system. In addition, the quantitative analysis clearly shows the bio-hallmarks such as wound closure, angiogenesis, blood flow, wound sealing and so on. Therefore, the manuscript is well organized and strongly recommended to be published with minor revisions.

Comments

1. Please explain in a little more detail why various bio-hallmarks should be observed simultaneously in the process of healing a wound and their importance.
2. In figure 3, the data is missed. Please cover the blank. If author cannot cover the image, please explain why the data is missing.
3. The authors mentioned the typical FOV in the studies were $> 2.5 \times 3 \text{ mm}^2$. What is the total imaging time to acquire 5 images in the typical FOV?

Reviewer #2 (Remarks to the Author):

This work reports a 5-modal intravital microscopy technique, which integrates optoacoustic (OA) microscopy, two-photon fluorescence microscopy, second-harmonic generation microscopy, third-harmonic generation microscopy, and bright-field light microscopy. Chronic wound healing is studied with the 5-modal microscopy in a label-free manner, showing the spatiotemporal interplay between wound closure, arteriogenesis, angiogenesis, and epithelial reformation.

Reporting a first-of-a-kind multi-modal microscopy for label-free intravital imaging, the technical innovation is high. Providing new insights into the cellular/molecular mechanisms underlying wound healing, the biological significance is also considerable. Thus, I suggest the publication of this work providing that the following comments are fully addressed.

1. Optoacoustic microscopy provides 3D images with 2D raster scan. While, nonlinear light microscopy (e.g., TPM, SHG, THG) required 3D scanning. It is not clear from the manuscript that if z-stack imaging is performed for TPM/SHG/THG. If so, it would be nice to show the co-registered 3D rendering of different contrasts. If not, where does the 300-micron penetration come from?
2. The lateral and axial resolutions of OA and nonlinear light microscopy are matched (~ 1 micron laterally and ~ 6 micron axially). However, the images shown in Fig. 2 look much blurred and do not reflect the micron-level high spatial resolution. Why?
3. Fig. b.3 is missing.
4. The configuration of the 5-modal microscope (i.e., transmission mode) limits its application to thin tissues. Please discuss it as a limitation in the manuscript.

Reviewer #3 (Remarks to the Author):

In this manuscript, Seeger et al. present a multi-modal microscopy platform optimized for label-free in-vivo imaging of wound healing hallmarks in mice. In particular, this platform, termed Co5M, offers concurrent recording of optoacoustic, 2P excited fluorescence, higher-harmonic as well as brightfield contrasts. After characterizing its performance in terms of spatial resolution, the authors then apply their technique to quantitatively monitor the process of wound healing in mice.

The multi-modal and label-free contrast enables them to analyze the complex spatial and temporal progression of several important hallmarks such as vessel dilation, angiogenesis, and general tissue remodeling.

Intravital microscopy is a quintessential tool to study dynamic cellular processes in-vivo and in the context of the natural tissue environment. Here the development by Seeger et al. is an important addition to the biomedical imaging toolbox, as it combines, for the first time, multiple label-free imaging modalities in a single platform. While common intravital recordings are typically restricted to a single or a couple of molecular reporters, many of which require the use of artificial labeling techniques, the study of Seeger et al. strikingly demonstrates the added advantage and potential of label-free contrast mechanism to disentangle complex spatio-temporal patterns of tissue remodeling.

The paper is, for most parts, clearly written, and the chosen application nicely demonstrates the technique's potential. Thus I am supportive of its publication in principle, however, I suggest the authors to clarify a couple of points below, which I hope will also strengthen the presentation of the paper with respect to the broad readership of Communications Biology.

Major comments/questions:

*) The authors have invested quite some effort into, and correspondingly highlight, the capability of the Co5M platform to 'concurrently' image the multiple modalities. However, since the temporal timescales of the processes under investigation span multiple hours to entire weeks, I am wondering how critical the (sub-second) concurrency really is here. While I appreciate the technical finesse required to synchronise the data acquisition over multiple detectors/modalities (see Fig. 1 and Methods), is it truly necessary? If the authors wish to further highlight this fact, maybe a more striking example demonstration of a highly dynamic process could be included. At minimum, this point should be openly discussed.

*) A second main comment relates to the origin of the label-free contrasts. The authors state that e.g. SHG primarily relates to tissue collagen, THG to lipid accumulations, autofluorescence emanates from elastin, etc. but in principle, I could imagine this to come from other tissue components as well. How have the authors ascertained their interpretation of the contrast origins?

*) The coupling of optical and acoustic modalities comes with some limitations regarding sample mounting and geometry, i.e. transducer and THG detector in transmission mode, acoustic coupling via water/gel, etc. I suggest the authors further discuss general limitations of their technique with respect to sample size and geometry. Eg. could Co5M also be used to study skin/wound healing in other tissue areas outside the (relatively thin) ear? What about more general future applications of Co5M?

Related to this, what is the potential influence of cross-talk between the modalities? I understand how detection is spectrally separated via filters (see SI Fig.1), but I could imagine that imperfect spectral filter properties could pose a problem?

*) The authors haven't motivated well why a Gaussian model and fit was chosen for the quantitative analysis presented in Fig. 4/5, and I believe the general reader would appreciate more details along this line. Also, how can the plateau of Fig. 5b be interpreted in terms of tissue composition, given that the authors state that past 60 days or so the wound healing process is completed.

Minor points:

*) If I understood correctly, the multi-photon and OA modalities are axially displaced from one another ($\sim 30\mu\text{m}$), owing to chromatic offsets. While this is being accounted for in post-processing (SI Fig. 3), couldn't this co-registration be rather easily fixed via optical alignment?

*) Line 160: How are the SNRs calculated for the modalities? Some additional data and/or uncertainties would be appreciated.

*) Fig.2: It would be helpful to indicate the wound site by a dashed circle or similar.

*) Fig. 3b, SI Fig 5 and 6 - some images appear missing.

*) Fig. 4: Labels are small and difficult to read

We want to thank the reviewers for the constructive feedback. We appreciate the thoughtful and positive comments, which have certainly helped to improve the presentation and quality of our paper. We have updated our paper according to the suggestions and performed more analysis as requested by the reviewers.

Answers to the reviewers in blue

Modifications of the manuscript in orange

Original text of the manuscript in green

Dear Professor Ntziachristos,

Your manuscript entitled "Label-free concurrent 5-modal microscopy (Co5M) resolves unknown spatio-temporal processes in wound healing" has now been seen by 3 referees. You will see from their comments below that while they find your work of considerable interest, some important points are raised. We are interested in the possibility of publishing your study in Communications Biology, but would like to consider your response to these concerns in the form of a revised manuscript before we make a final decision on publication.

We therefore invite you to revise and resubmit your manuscript, taking into account the points raised. In particular, as indicated by the reviewers, please include additional details and discuss the limitations of the proposed technique in the revised manuscript.

Please highlight all changes in the manuscript text file.

We are committed to providing a fair and constructive peer-review process. Do not hesitate to contact us if you wish to discuss the revision in more detail or if there are specific requests from the reviewers that you believe are technically impossible or unlikely to yield a meaningful outcome.

At the same time, we ask that you ensure your manuscript complies with our editorial policies. Please see our revision file checklist for guidance on formatting the manuscript and complying with our policies. You will also find guidelines for replying to the referees' comments. You may also wish to review our formatting guidelines for final submissions here.

Please use the following link to submit your revised manuscript, point-by-point response to the referees' comments (which should be in a separate document to the cover letter) and any additional files:

<https://mts-commsbio.nature.com/cgi-bin/main.plex?el=A3Cx6DCT4A4nsS3I4A9ftdsdWdWTJ3G69kvfuJVgbZ6gZ>

When submitting the revised version of your manuscript, please pay close attention to our Digital Image Integrity Guidelines and to the following points below:

- that unprocessed scans are clearly labelled and match the gels and western blots presented in figures.

- that control panels for gels and western blots are appropriately described as loading on sample processing controls
 - all images in the paper are checked for duplication of panels and for splicing of gel lanes.
- Finally, please ensure that you retain unprocessed data and metadata files after publication, ideally archiving data in perpetuity, as these may be requested during the peer review and production process or after publication if any issues arise.

We would expect revisions of this nature to take around three months, but appreciate that every situation is unique. We look forward to receiving your revised manuscript when it is ready, and will not enforce a hard deadline on this revision.

Please do not hesitate to contact me if you have any questions or would like to discuss these revisions further. We look forward to seeing the revised manuscript and thank you for the opportunity to review your work.

Best regards,

Chao Zhou, PhD
Editorial Board Member
Communications Biology
orcid.org/0000-0002-8679-3413

REVIEWER REPORT:

Referee expertise:

Referee #1: Biophotonics

Referee #2: Photoacoustic imaging

Referee #3: Biophotonics

COMMENTS TO AUTHOR:

Reviewer #1 (Remarks to the Author):

This manuscript, “Label-free concurrent 5-modal microscopy (co5M) resolves unknown spatio-temporal processes in wound healing”, presents a new concept of the co5M including optoacoustic, two-photon excitation fluorescence, second harmonic generation, third harmonic generation and bright field microscopy. The co5M can simultaneously monitored various metrics of the wound healing process in the mouse ear. The authors systematically summarized the imaging mechanism and well explained the role and performance of each system. In addition, the quantitative analysis clearly shows the bio-hallmarks such as wound closure, angiogenesis, blood flow, wound sealing and so on. Therefore, the manuscript is well organized and strongly recommended to be published with minor revisions.

Comments

1. Please explain in a little more detail why various bio-hallmarks should be observed simultaneously in the process of healing a wound and their importance.

We thank the reviewer for this comment. We initially shortened the Introduction Section and reasoned the importance and, subsequently, the monitoring of multiple bio-hallmarks in healing wounds in the Material and Methods (Wound Healing Model) section, see I. 584ff. Furthermore, we indicated Co5M’s capability to unravel the interplays in the Discussion at I. 406 *Co5M can, in the future, shed light onto unresolved interdependent processes including misregulated stem cell activation and vasculopathies (e.g. abnormal anastomoses^[1] or microangiopathy^[2-4]) or associated impaired epithelial growth and imbalanced tissue matrix maintenance (e.g. diabetic ulcers^[5] or scar formations^[6,7]) and associate them with underlying conditions, such as diabetes, sepsis, or infections.* Furthermore, we would like to draw the reviewer’s attention to the Introduction I. 64ff, in which we summarize current limitations of microscopy studies for healing wounds, especially pointing out the missing link between different bio-hallmarks greatly influencing each other.

To address the reviewers comment, we added the following paragraph to the Introduction at l. 64: *It is particularly important to elucidate the longitudinal responses of the tissue layers^[8-10], functional units and skin appendages during healing^[8-12] to resolve their mutual interdependency. Critical unknown orchestrations play a role in the interplay between misregulated stem cell activation and vasculopathies (e.g., abnormal anastomoses^[1] or microangiopathy^[2-4]) and associated impaired epithelial growth and imbalanced tissue matrix maintenance (e.g., diabetic ulcers^[5] or scar formations^[6,7]). It is currently challenging to unravel these effects and their association with underlying conditions, such as diabetes, sepsis, or infections: no method can reveal sufficient wealth of contrast to concurrently capture the disease-specific entanglement of bio-hallmarks involved in the healing process and lead to an accurate wound healing model.*

To avoid repetition, we rephrased the section in the Discussion at l. 406 to: *Co5M can, in the future, shed light onto unresolved interdependent processes including ~~misregulated stem cell activation and vasculopathies (e.g. abnormal anastomoses^[1] or microangiopathy^[2-4]) or associated impaired epithelial growth and imbalanced tissue matrix maintenance (e.g. diabetic ulcers^[5] or scar formations^[6,7])~~ and associate them with underlying conditions, such as diabetes, sepsis, or infections.*

2. In figure 3, the data is missed. Please cover the blank. If author cannot cover the image, please explain why the data is missing.

In order to ensure a healthy status of the mouse to survive the anesthesia, we monitored its physiological condition (see l. 607ff). We found that, especially when imaging right after inflicting the wound, the mouse's physiological parameters became unstable after about 40 min (very low breathing and heart rate). As the photo of each imaging session was taken after the multimodal microscopic raster scan and the brightfield microscopic scan, we aborted the imaging to prevent the need for potential euthanasia due to prolonged anesthesia, especially at the 1h-post-infliction measurement stressing the animal.

We added the following explanation to the manuscript in both the results section as well as the material and methods:

l. 248: *(..) or photographs (see Fig. 3a-i.3; b.3 is missing to shorten the duration of anesthesia to ensure survival of the mice)*

l. 541: *(..) achieved by laser-scanning only. The imaging protocol consisted of (1) anesthetizing the mouse and waiting for the mouse's physiological conditions to stabilize (~2 min); (2) positioning and preparing the ear for imaging (see Material and Methods – Animal Model) (~2 min); (3) aligning the FOV and adjusting the optical focusing (~ 1 min); (4) acquiring the simultaneous multimodal imaging (~50-60 min); (5) acquiring the brightfield microscope images (~5 min); (6) capturing a photo of the ear (~0.5 min); disassembling the holder (~2 min); (7) watching the mouse recover from the anesthesia (>5 min). The total duration of anesthesia was therefore below 1h 20 min. For the imaging 1h post wound infliction (see Fig. 3b.1-3, Supplementary Fig. 5b.1-3, and Supplementary Fig. 6b.1-3), the anesthesia was reduced to 1h by capturing a slightly smaller FOV and by omitting the photograph to ensure survival of the mouse.*

3. The authors mentioned the typical FOV in the studies were > 2.5 x 3 mm². What is the total imaging time to acquire 5 images in the typical FOV?

We thank the reviewer for this valuable comment. The imaging speed for the multimodal capturing with the herein applied parameters (number of pixels and averaging) was 7 min per mm² (see l. 204). This led to an image duration of <1h for the multimodal images. The brightfield microscope images were recorded in an automatic stage scanning mode within ~5 min.

We added the following lines to the Material and Methods section which also captures the reviewer's previous comment of missing data:

l. 541: (...) achieved by laser-scanning only. The imaging protocol consisted of (1) anesthetizing the mouse and waiting for the mouse's physiological conditions to stabilize (~2 min); (2) positioning and preparing the ear for imaging (see Material and Methods – Animal Model) (~2 min); (3) aligning the FOV and adjusting the optical focusing (~ 1 min); (4) acquiring the simultaneous multimodal imaging (~50-60 min); (5) acquiring the brightfield microscope images (~5 min); (6) capturing a photo of the ear (~0.5 min); disassembling the holder (~2 min); (7) watching the mouse recover from the anesthesia (>5 min). The total duration of anesthesia was therefore below 1h 20 min. For the imaging 1h post wound infliction (see Fig. 3b.1-3, Supplementary Fig. 5b.1-3, and Supplementary Fig. 6b.1-3), the anesthesia was reduced to 1h by capturing a slightly smaller FOV and by omitting the photograph to ensure survival of the mouse.

Reviewer #2 (Remarks to the Author):

This work reports a 5-modal intravital microscopy technique, which integrates optoacoustic (OA) microscopy, two-photon fluorescence microscopy, second-harmonic generation microscopy, third-harmonic generation microscopy, and bright-field light microscopy. Chronic wound healing is studied with the 5-modal microscopy in a label-free manner, showing the spatiotemporal interplay between wound closure, arteriogenesis, angiogenesis, and epithelial reformation.

Reporting a first-of-a-kind multi-modal microscopy for label-free intravital imaging, the technical innovation is high. Providing new insights into the cellular/molecular mechanisms underlying wound healing, the biological significance is also considerable. Thus, I suggest the publication of this work providing that the following comments are fully addressed.

1. Optoacoustic microscopy provides 3D images with 2D raster scan. While, nonlinear light microscopy (e.g., TPM, SHG, THG) required 3D scanning. It is not clear from the manuscript that if z-stack imaging is performed for TPM/SHG/THG. If so, it would be nice to show the co-registered 3D rendering of different contrasts. If not, where does the 300-micron penetration come from?

We thank the reviewer for this valuable comment and would like to draw his/her attention to the Supplementary Fig. 3 showing the 3D co-registration after chromatic aberration correction of a black suture phantom generating OA and THG contrast.

During the main imaging, no z-stack imaging for the multiphoton modalities was performed, which would increase the imaging duration significantly. As described in l. 168, we utilized the chromatic aberration between the optoacoustic and the multiphoton excitation focus to position them in the respective depths within the mouse ear tissue. As reported in ^[13-16], the mouse ear's outer areas typically range between 350 and 450 μm in thickness and consist of a central cartilage sheet ($\sim 50 \mu\text{m}$) bordered on each side by $\sim 100 \mu\text{m}$ thick layers of subcutaneous tissue, dermis, and epidermis (inner to outer). Vasculature structures are typically located in the dermis/hypodermis and, thus, at depths of $150 \pm 50 \mu\text{m}$, and the connective tissue and sebaceous glands are located in the epidermis at depths of $50 \pm 50 \mu\text{m}$.

We added following lines to the Material and Methods section: l.632 (..) *of the extracellular matrix. The thickness of these tissue layers typically ranges about $\sim 100 \mu\text{m}$ for the epidermis containing the connective tissue and sebaceous glands, $\sim 100 \mu\text{m}$ for the dermis/hypodermis containing shallower microvasculature and deeper bigger vasculature structures*^[13-16].

To appropriately image the respective tissue layers, an imaging depth of 250-300 μm is thus required. We added the following line to the manuscript: l.193 (..) *An imaging depth of $>300 \mu\text{m}$ is achieved for all multiphoton and optoacoustic modes allowing to appropriately position the imaging planes within the respective tissue layers containing vasculature, connective tissue, and sebaceous glands (see Material and Methods)*^[13-16].

2. The lateral and axial resolutions of OA and nonlinear light microscopy are matched ($\sim 1 \mu\text{m}$ laterally and $\sim 6 \mu\text{m}$ axially). However, the images shown in Fig. 2 look much blurred and do not reflect the micron-level high spatial resolution. Why?

We thank the reviewer for pointing out a seemingly blurred image resolution. The lateral and axial resolution of Co5M was, as commonly performed, determined using miniature beads (see Supplementary Fig. 2). Such characterization demonstrates the best-achievable resolution within perfect sample conditions and does not reflect the resolution achievable in biological tissue, which is compromised by scattering as well as miniature movements of the mice.

However, to highlight the resolution achieved in vivo, we would like to draw the reviewer's attention to the zoom-in panels of Fig. 2, in which microstructures such as microcapillaries (typically ranging below $<5 \mu\text{m}$ [15,17-19]) and collagen fibers (typically ranging below $<20 \mu\text{m}$ [20-23]) can be seen. We added the following image to the Supplementary information to demonstrate that Co5M revealed microcapillaries and collagen fibers below $9 \mu\text{m}$ (Microcapillary: $7.61 \mu\text{m}$ (FWHM), $R: 0.9619$; Collagen Fiber: $8.10 \mu\text{m}$ (FWHM), $R: 0.9205$).

Supplementary Figure 7 Microstructures of tissue imaged with Co5M in vivo using the OA (day 14) and SHG (day 4) images of mouse #1. (a) Zoom-in mouse ear vasculature imaged with OA and (b) Line profile of microvessel indicated with a red line in (a) with a diameter of 7.61 (Gaussian Fit, FWHM, $R=0.9619$). (c) Zoom-in Mouse ear collagen network imaged with SHG and (d) Line profile of collagen fibril indicated with a red line in (c) with a diameter of 8.10 (Gaussian Fit, FWHM, $R=0.9205$).

We further analyzed the resolution of our multimodal images using Fourier Shell Correlation for the OA and SHG images. We used the SHG images as representatives for all equipped multiphoton modalities as

they contain the finest spatial structures (collagen fibrils) and the resolution of 2PEF and THG can be assumed identical (see Supplementary Fig. 2).

We changed and added the following lines to the manuscript to provide an explanation for the reduced resolution at l. 183: *Co5M achieved imaging resolutions of $\sim 1.1 \mu\text{m}$ laterally and $\sim 5.7 \mu\text{m}$ axially for the SHG, 2PEF, and THG mode, and $\sim 1.5 \mu\text{m}$ laterally for BF (see Supplementary Fig. 2 and 3). OA resolution was $\sim 0.6 \mu\text{m}$ laterally and $\sim 6.3 \mu\text{m}$ axially, owing to a diffraction limited optical excitation and a highest detected frequency of 110 MHz. The spatial resolution of Co5M is compromised when imaging biological tissue in vivo due to light scattering as well as microscopic movements of the living tissue to within $\sim 2.8 \mu\text{m}$ determined by Fourier Shell Correlation (see Supplementary Fig. 8)^[24–26], which allows Co5M to image biological microstructures below $8 \mu\text{m}$ in diameter such as microcapillaries (typically ranging below $<10 \mu\text{m}$ ^[15,17–19]) and collagen fibrils (typically ranging below $<20 \mu\text{m}$ ^[20–23]) (see Supplementary Fig. 7).*

We further added the

following figure to the Supplementary Information:

a.

b.

c.

d.

3. Fig. b.3 is missing.

Please see comment 2 of reviewer #1.

4. The configuration of the 5-modal microscope (i.e., transmission mode) limits its application to thin tissues. Please discuss it as a limitation in the manuscript.

We thank the reviewer for pointing out the maximum sample thickness Co5M allows due to its transmission mode configuration. We added the following lines to the Discussion: l. 408 (..) *nearly any process in accessible tissue that is characterized by the key biological functional units discussed here. Whereas the current configuration of Co5M allows a maximum sample thickness of <500 μm due to its transmission mode operation, first steps were taken towards redesigning the interrogation area for epi-mode imaging^[27–30], facilitating the investigation of thicker specimen. To counteract the inherent maximum imaging depth of $\sim 300 \mu\text{m}$ for the optical resolution microscopy modalities used herein, multiscale imaging methods and approaches, especially for the optoacoustic mode, would allow to fuse microscopic with mesoscopic findings^[31].*

Reviewer #3 (Remarks to the Author):

In this manuscript, Seeger et al. present a multi-modal microscopy platform optimized for label-free in-vivo imaging of wound healing hallmarks in mice. In particular, this platform, termed Co5M, offers concurrent recording of optoacoustic, 2P excited fluorescence, higher-harmonic as well as brightfield contrasts. After characterizing its performance in terms of spatial resolution, the authors then apply their technique to quantitatively monitor the process of wound healing in mice. The multi-modal and label-free contrast enables them to analyze the complex spatial and temporal progression of several

Supplementary Figure 8 Resolution determination of Co5M mouse tissue imaging in vivo using the OA and SHG images of mouse #2 at day 14. (a) OA image of microvasculature and corresponding (b) Fourier shell correlation with μm -scaled pixels reveal a Fourier Image Resolution (FIRE) of $2.773 \mu\text{m}$. (c) SHG image of extracellular collagen fibers and (d) Fourier shell correlation with μm -scaled pixels reveal a FIRE of $2.913 \mu\text{m}$.

important hallmarks such as vessel dilation, angiogenesis, and general tissue remodeling.

Intravital microscopy is a quintessential tool to study dynamic cellular processes in-vivo and in the context of the natural tissue environment. Here the development by Seeger et al. is an important addition to the biomedical imaging toolbox, as it combines, for the first time, multiple label-free imaging modalities in a single platform. While common intravital recordings are typically restricted to a single or a couple of molecular reporters, many of which require the use of artificial labeling techniques, the study of Seeger et al. strikingly demonstrates the added advantage and potential of label-free contrast mechanism to disentangle complex spatio-temporal patterns of tissue remodeling.

The paper is, for most parts, clearly written, and the chosen application nicely demonstrates the technique's potential. Thus I am supportive of its publication in principle, however, I suggest the authors to clarify a couple of points below, which I hope will also strengthen the presentation of the paper with respect to the broad readership of Communications Biology.

Major comments/questions:

*) The authors have invested quite some effort into, and correspondingly highlight, the capability of the Co5M platform to 'concurrently' image the multiple modalities. However, since the temporal timescales of the processes under investigation span multiple hours to entire weeks, I am wondering how critical the (sub-second) concurrency really is here. While I appreciate the technical finesse required to synchronise the data acquisition over multiple detectors/modalities (see Fig. 1 and Methods), is it truly necessary? If the authors wish to further highlight this fact, maybe a more striking example demonstration of a highly dynamic process could be included. At minimum, this point should be openly discussed.

We thank the reviewer for this valuable comment. We agree that the herein monitored process does not necessarily require the synchronization to have a precision on the millisecond scale. However, in order to appropriately monitor healing bio-hallmarks, a rather large FOV is required (min 2 x 2 mm²). Considering the image acquisition of the OA mode to be about as fast as the MP modes, sequential imaging of such an area would 1) double the acquisition time, compromising the maximum allowed duration of mouse anesthesia in pre-clinical settings (< 1h 30min), or 2) half the FOV, compromising the necessary area to be imaged (> 2 x 2 mm²). Hence, only concurrent/simultaneous multimodal acquisition allows for appropriate imaging of wound healing. To achieve such truly simultaneous multimodal acquisition, any crosstalk among the modalities has to be reduced as much as possible, rendering the need for a precise triggering scheme.

We would like to draw the reviewers attention to the explanation given in the manuscript: l.136ff *The synchronization of all equipped devices (i.e., lasers, detectors, scanning units, and data acquisition cards) by a system-wide shared triggering scheme and a time-locked data acquisition allows simultaneous recording of the modalities with negligible crosstalk. In combination with a fully-automated scanning procedure based on fusing laser- and sample-scanning concepts, Co5M can now obtain high-fidelity images of key tissue components with micrometer resolution across millimeter-sized areas within preclinical time-frames, enabling the extraction of quantitative metrics that describe aspects of biological processes such as wound healing.*

To further give information about the time-limiting factors of pre-clinical imaging of healing wounds, we added the following lines to the Material and Methods section:

l. 541: (...) achieved by laser-scanning only. The imaging protocol consisted of (1) anesthetizing the mouse and waiting for the mouse's physiological conditions to stabilize (~2 min); (2) positioning and preparing the ear for imaging (see Material and Methods – Animal Model) (~2 min); (3) aligning the FOV and adjusting the optical focusing (~ 1 min); (4) acquiring the simultaneous multimodal imaging (~50-60 min); (5) acquiring the brightfield microscope images (~5 min); (6) capturing a photo of the ear (~0.5 min); disassembling the holder (~2 min); (7) watching the mouse recover from the anesthesia (>5 min). The total duration of anesthesia was therefore below 1h 20 min. For the imaging 1h post wound infliction (see Fig. 3b.1-3, Supplementary Fig. 5b.1-3, and Supplementary Fig. 6b.1-3), the anesthesia was reduced to 1h by capturing a slightly smaller FOV and by omitting the photograph to ensure survival of the mouse.

*) A second main comment relates to the origin of the label-free contrasts. The authors state that e.g. SHG primarily relates to tissue collagen, THG to lipid accumulations, autofluorescence emanates from

elastin, etc. but in principle, I could imagine this to come from other tissue components as well. How have the authors ascertained their interpretation of the contrast origins?

We thank the reviewer for pointing out potential misinterpretations of the signals' origins. In general, especially when imaging biological tissue, all imaging modalities detect intrinsic background signals as well as other detectable signals from secondary biological moieties. However, the main signal contribution sources for the herein implemented modalities on unstained mammalian tissue are well known and specific:

- OA: Optoacoustics is based on intrinsic absorption of tissue chromophores with high abundance, high local concentration, high absorption coefficient, and negligible quantum yield. When applied in the visible range between 500-650 nm, the two most important tissue chromophores of mammalian tissue are melanin and hemoglobin. The herein used animal model of an athymic nude Foxn1 lacks melanin, which indicates that most of the recorded signals originate from hemoglobin stored in red blood cells within the vasculature network. Other tissue light absorbing chromophores with relevant concentrations and, thus, OA contrast in athymic nude Foxn1 animals are lipids and water. However, when comparing their absorption coefficient at the employed excitation wavelength, we could neglect their contribution to the recorded OA signals^[32-36].
- SHG: Second Harmonic generation relies on a non-vanishing second order susceptibility tensor, which in general terms requires optical birefringence. Such optical birefringing materials are typically based on non-centrosymmetric crystals (e.g., Nd:YAG) employed for frequency doubling in laser systems. In biological materials, birefringing materials with non-centrosymmetry undergoing optical second harmonic generation, are elongated structures without inversion symmetry (collagen type I, II, IV, myosin, and to some extent elastin) as well as bone. Considering the herein used model for wound healing, we do not expect significant myosin (muscle fibers) or bone tissue in the mouse ear. Hence, we can assume that almost all SHG is generated by the extracellular collagen matrix^[37-42].
- THG: Third Harmonic Generation is based on optical interfaces defined as a rapid change of refractive index within the spatial extent of the optical focus, i.e. a non-vanishing third order susceptibility tensor. This being said, THG in general is able to sense and detect all optical boundaries. In biological tissue, the strongest interfaces of different refractive indices occur when lipid-rich and water-rich areas meet. Such interfaces are therefore either lipid bilayers (cell membranes, especially keratinocytes of the epidermis) or local lipid accumulations such as the one in skin appendices such as sebaceous glands^[37-40,42].
- 2PEF: Two-photon excitation fluorescence based on tissue autofluorescence relies on intrinsic absorption and non-vanishing quantum yield. Whereas a huge variety of tissue chromophores contributes to tissue autofluorescence, the most dominant ones when exciting melanin-free mammalian tissue at 1043 nm (i.e. 521.5 nm for 2PE) are elastin, FAD, NADH, and keratin^[38,41-43].

Besides the underlying physical considerations of the modalities of Co5M being rather selective for the herein demonstrated wound healing study in mouse ears using an athymic nude Foxn1 animal model, our analysis is not based on the signal amplitude but rather the spatial structure. As an example, whereas OA could originate from other light absorbing moieties within the mouse ear, our analysis is based on characterizing the vasculature tree in its degree of branching as well as vessel thickness. Analogously, the THG modalities are not used to determine or measure the lipid concentration within the tissue but 'only' for localizing sebaceous glands qualitatively and then track these skin appendages over time and space.

To clarify this approach, we added the following lines to the manuscript: l.130 (..) *microscopy for providing an anatomical reference. For the herein used athymic nude Hsd Foxn1 mouse model, we expect Co5M's OA signals to be dominated by hemoglobin^[32-36], SHG signals by the extracellular collagen matrix^[37-42], THG signals by lipid bilayers (cell membranes, especially keratinocytes of the epidermis) and local lipid accumulations such as in sebaceous glands^[37-40,42], and 2PEF signals by elastin, FAD, NADH, and keratin^[38,41-43]. Co5M's modalities access that by intrinsic biological contrast, which is analyzed to outline the spatial characteristics of the observed biological moieties.*

*) The coupling of optical and acoustic modalities comes with some limitations regarding sample mounting and geometry, i.e. transducer and THG detector in transmission mode, acoustic coupling via water/gel, etc. I suggest the authors further discuss general limitations of their technique with respect to sample size and geometry. Eg. could Co5M also be used to study skin/wound healing in other tissue areas outside the (relatively thin) ear? What about more general future applications of Co5M? Related to this, what is the potential influence of cross-talk between the modalities? I understand how detection is spectrally separated via filters (see SI Fig.1), but I could imagine that imperfect spectral filter properties could pose a problem?

We thank the reviewer for pointing out the current limitations of Co5M regarding sample thickness caused by its transmission mode configuration. We would like to draw his/her attention to comment 4 of reviewer 2, where we have given an explanation and manuscript modification regarding the sample thickness.

Regarding future augmentations of the working principles of Co5M, we would like to highlight our statement and changes in the Discussion: l. 406 (..) *Co5M's combination of simultaneous, multimodal, and label-free capabilities, which are applicable across a large FOV with sub-cellular precision and high SNR, makes it a versatile platform for longitudinal studies of nearly any process in accessible tissue that is characterized by the key biological functional units discussed here. Whereas the current configuration of Co5M allows a maximum sample thickness of <500 μm due to its transmission mode operation, first steps were taken towards redesigning the interrogation area for epi-mode imaging^[27-30], facilitating the investigation of thicker specimen. To counteract the inherent maximum imaging depth of $\sim 300 \mu\text{m}$ for the optical resolution microscopy modalities used herein, multiscale imaging methods and approaches, especially for the optoacoustic mode, would allow to fuse microscopic with mesoscopic findings^[31].*

As explained in the previous point raised by the reviewer, the data/images of Co5M for this study are analyzed in their spatial characteristics. Potential crosstalk between the modalities thus has only a minor effect on e.g. the spatial structure and extent of the vasculature network. Furthermore, the modalities are based on different working principles: The OA modality can be assumed unharmed/uninfluenced by the MP laser, as that laser's repetition rate is $\sim 80 \text{ MHz}$ and, thus, would only contribute to the recorded OA signals as a single frequency component (^[44]) of low amplitude; the optical modalities can be assumed unharmed/uninfluenced by the OA laser due to the specific time-locked acquisition shifting the optical recording window of 2PEF, SHG, and THG by 10 μs to the OA laser pulse. Considering the slowest optical processes (lifetime of the first excited state relevant for fluorescent processes) in the order of $< 20 \text{ ns}$, a negligible contribution of fluorescence excitation with the OA laser can be assumed. The crosstalk among the pure optical modalities (SHG, THG, and 2PEF), as for every multiphoton microscope, cannot be fully neglected due to imperfections in the optics. As standardly applied in the field, optical dichroics, notch-, or bandpass filters are used to minimize the crosstalk.

Therefore, the combination of physically different underlying processes leading to signal generation, the lack of other influencing chromophores in the herein used mouse model, the time-locked data

acquisition, and the analysis focusing on spatial characteristics instead of intensity values, all ensure that no crosstalk among the modalities can occur and influence the analysis.

*) The authors haven't motivated well why a Gaussian model and fit was chosen for the quantitative analysis presented in Fig. 4/5, and I believe the general reader would appreciate more details along this line. Also, how can the plateau of Fig. 5b be interpreted in terms of tissue composition, given that the authors state that past 60 days or so the wound healing process is completed.

We thank the reviewer for pointing the attention to the applied mathematical model. We decided to utilize Gaussian models because of two reasons: first, a Gaussian model is simple and easy to grasp and, second, our data shows a spatial characteristic of hallmarks responding to a stimulus (i.e., the wound) so we expect their reaction to follow a spatial distribution typically described using Gaussian models. Furthermore, we decided on using simple Gaussian models to not induce a bias to the results of our data when applying more complex models.

Regarding the long-lasting plateau phase of the monitored hallmarks, we would like to draw the reviewer's attention to the explanation in l. 318, which we further modified to give more information: *All metrics enter a long-lasting plateau phase, indicating that the associated hallmarks approach, but never reach, their original condition before injury. Despite the healing process considered as completed, a long-lasting difference of the monitored hallmarks when compared to the before-image can be interpreted as very mild scarring^[45,46].*

Minor points:

*) If I understood correctly, the multi-photon and OA modalities are axially displaced from one another (~30µm), owing to chromatic offsets. While this is being accounted for in post-processing (SI Fig. 3), couldn't this co-registration be rather easily fixed via optical alignment?

The chromatic aberration is an intrinsic property of diffractive optics, like the herein used microscopic objective lens, to focus light of a specific wavelength. The different focal lengths of e.g. 532 nm and 1043 nm (the two excitation sources used in this study) experience different refractive indices of the lens-air-interface and, thus, are diffracted wavelength-dependently. Hence, the best achievable optical focusing of the given wavelengths, which requires perfectly collimated beams encountering the back aperture of the objective lens, are focused to different planes.

In order to counteract the chromatic offsets, re-alignment of the optical system would lead to inappropriate optical focus. A slightly diverging or converging beam entering the back aperture of the objective lens would impede the diffraction limited focusing, induce artifacts, and, thus, alter the achieved optical resolution.

Reflective optics could solve the issue of chromatic aberrations as the focal length is wavelength-independent. In that case, all applied wavelengths are focused to the identical focal plane. However, in this study, the chromatic aberrations were of benefit as the different modalities target biological moieties at different depths (e.g. vasculature in dermis, connective tissue in epidermis). We added the following line to the manuscript to explain this potential augmentation of Co5M for imaging targets where no chromatic aberrations would be of benefit: l. 415. *Furthermore, whereas the chromatic aberration of the applied excitation wavelengths was beneficial to image various biological moieties at different depths within the skin, implementing reflective focusing elements could allow Co5M to apply multimodal microscopy on a single-cell level without refocusing or post-processing.*

*) Line 160: How are the SNRs calculated for the modalities? Some additional data and/or uncertainties would be appreciated.

The SNRs of Co5M's modalities were characterized as a function of the applied pulse energy in the focus of the respective laser. Multimodal images of in vivo mouse ears were acquired at pulse energies of ~53 nJ (OA) and 3.6 nJ (MP), and 50 averages. The noise levels when blocking the laser beams were found at 7.9 mV (OA), 0.7 mV (2PEF and SHG), and 7.2 mV (THG). The THG noise level was ~10x higher than the other MP modalities as the used optical signal amplifier applied a 10^6 amplification instead of a 10^5 amplification used for 2PEF and SHG due to its intrinsic lower intensity as well as reduced sensitivity of the equipped PMTs at the THG wavelength. The SNRs were calculated using the averaged signal thresholded above the respective background level with $SNR = 20 \cdot \log(\text{Signal/Noise})$ and led to SNRs of 33.4 dB (OA), 36.4 dB (2PEF), 44.8 dB (SHG), and 27.2 dB (THG).

*) Fig.2: It would be helpful to indicate the wound site by a dashed circle or similar.

We would like to refuse adding a dashed line outlining the wound of the 1h-post infliction image to the relevant images as we believe it covers interesting features not visible anymore to a reader. As an example, in the zoom-ins of Fig. 2, a dashed outline of the wound hides the newly formed tortuous microcapillaries.

*) Fig. 3b, SI Fig 5 and 6 - some images appear missing.

Please see comment 2 of reviewer #1.

*) Fig. 4: Labels are small and difficult to read

We changed the font size of Fig. 4

References

- [1] M. K. Schneider, H. I. Ioanas, J. Xandry, M. Rudin, *Sci. Rep.* **2019**, *9*, 1.
- [2] H. Konya, *World J. Diabetes* **2014**, *5*, 678.
- [3] R. D. Galiano, O. M. Tepper, C. R. Pelo, K. A. Bhatt, M. Callaghan, N. Bastidas, S. Bunting, H. G. Steinmetz, G. C. Gurtner, *Am. J. Pathol.* **2004**, *164*, 1935.
- [4] K. E. Johnson, T. A. Wilgus, *Adv. Wound Care* **2014**, *3*, 647.
- [5] L. Braiman-Wiksmann, I. Solomonik, R. Spira, T. Tennenbaum, *Toxicol. Pathol.* **2007**, *35*, 767.
- [6] S. M. McCarty, C. A. Cochrane, P. D. Clegg, S. L. Percival, *The role of endogenous and exogenous enzymes in chronic wounds: A focus on the implications of aberrant levels of both host and bacterial proteases in wound healing*, Vol. 20, **2012**, pp. 125–136.
- [7] H. Liu, H. Liu, X. Deng, M. Chen, X. Han, W. Yan, N. Wang, *Anal. Methods* **2016**, *8*, 3503.
- [8] G. Deka, S.-W. Chu, F.-J. Kao, In *Microscopy and Analysis*, InTech, **2016**, p. 16.
- [9] P. Martin, *Science (80-.)*. **1997**, *276*, 75.
- [10] A. H. Zhou, *Int. J. Intell. Control Syst.* **2012**, *17*, 79.
- [11] S. Dekoninck, C. Blanpain, *Nat. Cell Biol.* **2019**, *21*, 18.
- [12] M. Aragona, S. Dekoninck, S. Rulands, S. Lenglez, G. Mascré, B. D. Simons, C. Blanpain, *Nat. Commun.* **2017**, *8*.
- [13] J. H. Barker, D. Kjolseth, M. Kim, J. M. Frank, I. Bondar, E. Uhl, M. Kamler, K. Messmer, G. R. Tobin, L. J. Weiner, *Wound Repair Regen.* **1994**, *2*, 138.
- [14] S. Yousefi, J. Qin, S. Dziennis, R. K. Wang, *J. Biomed. Opt.* **2014**, *19*, 076015.
- [15] J. H. Barker, F. Hammersen, I. Bondar, E. Uhl, T. Galla, M. Menger, K. Messmer, **1989**, 948.
- [16] H. D. Zomer, A. G. Trentin, *J. Dermatol. Sci.* **2018**, *90*, 3.
- [17] N. Honkura, M. Richards, B. Laviña, M. Sáinz-Jaspeado, C. Betsholtz, L. Claesson-Welsh, *Nat. Commun.* **2018**, *9*.
- [18] M. Seeger, D. Soliman, J. Aguirre, G. Diot, J. Wierzbowski, V. Ntziachristos, *Nat. Commun.* **2020**, *11*, 2910.
- [19] D. C. Chong, Z. Yu, H. E. Brighton, J. E. Bear, V. L. Bautch, *Arterioscler. Thromb. Vasc. Biol.* **2017**, *37*, 1903.
- [20] S. Bancelin, C. Aimé, I. Gusachenko, L. Kowalczyk, G. Latour, T. Coradin, M. C. Schanne-Klein, *Nat. Commun.* **2014**, *5*, 1.
- [21] T. USHIKI, *Arch. Histol. Cytol.* **2002**, *65*, 109.
- [22] C. G. Fuentes-Corona, J. Licea-Rodriguez, R. Younger, R. Rangel-Rojo, E. O. Potma, I. Rocha-Mendoza, *Biomed. Opt. Express* **2019**, *10*, 6449.
- [23] X. Chen, O. Nadiarynk, S. Plotnikov, P. J. Campagnola, *Nat. Protoc.* **2012**, *7*, 654.
- [24] R. P. J. Nieuwenhuizen, K. A. Lidke, M. Bates, D. L. Puig, D. Grünwald, S. Stallinga, B. Rieger, *Nat. Methods* **2013**, *10*, 557.
- [25] M. Van Heel, M. Schatz, *J. Struct. Biol.* **2005**, *151*, 250.

- [26] S. Koho, G. Tortarolo, M. Castello, T. Deguchi, A. Diaspro, G. Vicidomini, *Nat. Commun.* **2019**, *10*.
- [27] R. Shnaiderman, G. Wissmeyer, M. Seeger, D. Soliman, H. Estrada, D. Razansky, A. Rosenthal, V. Ntziachristos, *Optica* **2017**, *4*, 1180.
- [28] R. Shnaiderman, G. Wissmeyer, H. Estrada, M. Seeger, V. Ntziachristos, In *Photons Plus Ultrasound: Imaging and Sensing 2018* (Eds.: Oraevsky, A. A.; Wang, L. V.), SPIE, **2018**, p. 92.
- [29] G. Wissmeyer, D. Soliman, R. Shnaiderman, A. Rosenthal, V. Ntziachristos, *Opt. Lett.* **2016**, *41*, 1953.
- [30] G. Wissmeyer, R. Shnaiderman, D. Soliman, V. Ntziachristos, In *Photons Plus Ultrasound: Imaging and Sensing 2017* (Eds.: Oraevsky, A. A.; Wang, L. V.), International Society for Optics and Photonics, **2017**, p. 1006423.
- [31] D. Soliman, G. J. Tserevelakis, M. Omar, V. Ntziachristos, *Sci. Rep.* **2015**, *5*, 1.
- [32] V. Ntziachristos, *Nat. Methods* **2010**, *7*, 603.
- [33] H. Zafar, M. Leahy, W. Wijns, M. C. Kolios, J. Zafar, N. Johnson, F. Sharif, *Biomed. Phys. Eng. Express* **2018**, *4*.
- [34] J. Weber, P. C. Beard, S. E. Bohndiek, *Contrast agents for molecular photoacoustic imaging*, Vol. 13, Nature Publishing Group, **2016**, pp. 639–650.
- [35] V. Ntziachristos, D. Razansky, *Chem. Rev.* **2010**, *110*, 2783.
- [36] C. Lutzweiler, D. Razansky, C. Lutzweiler, D. Razansky, *Sensors (Basel)*. **2013**, *13*, 7345.
- [37] P. C. Wu, T. Y. Hsieh, Z. U. Tsai, T. M. Liu, *Sci. Rep.* **2015**, *5*, 34.
- [38] H. Segawa, M. Okuno, H. Kano, P. Leproux, V. Couderc, H. Hamaguchi, *Opt. Express* **2012**, *20*, 9551.
- [39] B. Weigelin, G.-J. Bakker, P. Friedl, *J. Cell Sci.* **2016**, *c*, 245.
- [40] D. Sandkuijl, A. E. Tuer, D. Tokarz, J. E. Sipe, V. Barzda, *J. Opt. Soc. Am. B* **2013**, *30*, 382.
- [41] C. Stringari, L. Abdeladim, G. Malkinson, P. Mahou, X. Solinas, I. Lamarre, S. Brizion, J. B. Galey, W. Supatto, R. Legouis, A. M. Pena, E. Beaurepaire, *Sci. Rep.* **2017**, *7*, 1.
- [42] J. D. Jones, H. E. Ramser, A. E. Woessner, K. P. Quinn, *Commun. Biol.* **2018**, *1*.
- [43] A. J. Radosevich, B. Bouchard, Matthew, S. A. Burgess, B. R. Chen, M. C. Hillman, Elizabeth, **2009**, *8*, 425.
- [44] S. Kellnberger, D. Soliman, G. J. Tserevelakis, M. Seeger, H. Yang, A. Karlas, L. Prade, M. Omar, V. Ntziachristos, *Light Sci. Appl.* **2018**, *7*, 109.
- [45] B. Gawronska-Kozak, *Matrix Biol.* **2011**, *30*, 290.
- [46] J. Bukowska, M. Kopcewicz, K. Walendzik, B. Gawronska-Kozak, *Int. J. Mol. Sci.* **2018**, *19*, 19.

REVIEWERS' COMMENTS:

Reviewer #2 (Remarks to the Author):

I recommend the publication of this work providing that the following comments are addressed.

1. The authors should clarify in the manuscript that no z-stack imaging for the multiphoton modalities was performed.
2. The authors should include a 3D rendering of the ear vasculature imaged by PAM in the supplementary information to show that the 6 micron axial resolution can be achieved in vivo.

Reviewer #3 (Remarks to the Author):

I have carefully read the revised version of the manuscript as well as the point-by-point reply to my original comments. Overall I am satisfied with the responses and think the paper's presentation has improved in clarity, especially for a broad readership. Thus I recommend publication in the present form.

Robert Prevedel, EMBL Heidelberg